# UNDERSTANDING THE MIXTURE-OF-EXPERTS WITH NADARAYA-WATSON KERNEL

**Chuanyang Zheng**[1]* **Jiankai Sun**[2] **Yihang Gao**[3] **Enze Xie**[4] **Yuehao Wang**[5]
**Peihao Wang** [5] **Ting Xu**[6] **Matthew Chang**[7] **Liliang Ren**[8] **Jingyao Li**[6]
**Jing Xiong**[9] **Kashif Rasul**[1] **Mac Schwager**[2] **Anderson Schneider**[1] **Atlas Wang**[5]
**Yuriy Nevmyvaka**[1]
[1]Morgan Stanley [2]Stanford [3]NUS [4]Nvidia [5]UT Austin
[6]CUHK [7]Meta [8]Microsoft Research [9]HKU

## ABSTRACT

Mixture-of-Experts (MoE) has become a cornerstone in recent state-of-the-art large language models (LLMs). Traditionally, MoE relies on $\mathrm{Softmax}$ as the router score function to aggregate expert output, a designed choice that has persisted from the earliest MoE models to modern LLMs, and is now widely regarded as standard practice. However, the necessity of using $\mathrm{Softmax}$ to project router weights into a probability simplex remains an unchallenged assumption rather than a principled design choice. In this work, we first revisit the classical Nadaraya–Watson regression and observe that MoE shares the same mathematical formulation as Nadaraya–Watson regression. Furthermore, we show that both feed-forward neural network (FFN) and MoE can be interpreted as a special case of Nadaraya–Watson regression, where the kernel function corresponds to the input neurons of the output layer. Motivated by these insights, we propose the **zero-additional-cost** Kernel Inspired Router with Normalization (KERN), an FFN-style router function, as an alternative to $\mathrm{Softmax}$. We demonstrate that this router generalizes both Sigmoid- and Softmax-based routers. **Based on empirical observations and established practices in FFN implementation, we recommend the use of** $\mathrm{ReLU}$ **activation and** $\ell_2$**-normalization in** KERN **router function.** Comprehensive experiments in MoE and LLM validate the effectiveness of the proposed FFN-style router function KERN.

## 1 INTRODUCTION

Recent years have witnessed remarkable progress in Large Language Models (LLMs) (Brown et al., 2020; Ouyang et al., 2022; Touvron et al., 2023), driven primarily by the exponential growth of training data and model parameters. With the mixture of experts (MoE), there is great progress in language modeling (Fedus et al., 2022; Puigcerver et al., 2024; Jiang et al., 2024; Meta; Liu et al., 2024a; Team et al., 2025) and computer vision (Riquelme et al., 2021; Lin et al., 2024b). The MoE architecture (Jacobs et al., 1991; Shazeer et al., 2017; Roller et al., 2021) has emerged as an efficient alternative that allows parameter scaling while maintaining manageable computational requirements. The successful integration of MoE with Transformer architectures (Vaswani et al., 2017) has led to the development of exceptionally large yet efficient language models (Dai et al., 2024; Jiang et al., 2024; Shen et al., 2024; Wei et al., 2024; Liu et al., 2024a), demonstrating the tremendous potential of this approach.

A critical and widely adopted design choice in modern MoE architectures is the use of the $\mathrm{Softmax}$ function as the core routing mechanism. This approach, prominently featured in large-scale models (Lepikhin et al., 2021; Jiang et al., 2024; Liu et al., 2024a; Team et al., 2025), has effectively become the de facto standard for state-of-the-art systems. The function $\mathrm{Softmax}$ naturally induces a probability distribution on the available experts. This property ensures that the routing weights for each token sum to one, promoting a balanced and interpretable allocation. However, despite its prevalence and intuitive appeal, the theoretical justification for its exclusive dominance remains

---

*Contact Email: cyzhengme@gmail.com

somewhat unclear. Recently, Sigmoid has been proven to be a better router function (Nguyen et al., 2024a), which is also investigated and adopted as an alternative router score function by DeepSeek (Dai et al., 2024; Liu et al., 2024a). Their findings suggest that a Sigmoid-based routing function performs effectively in MoE.

In this work, we revisit the fundamental design principles of MoE routing by re-examining it through the statistical lens of the Nadaraya-Watson regression estimator (Nadaraya, 1964; Watson, 1964). We propose a novel interpretation: the router's output for a given input token can be viewed as a set of dynamic kernel weights assigned to each expert. Each expert, in turn, acts as a value function, producing an aggregated output. This perspective is further reinforced by the architectural parallels within the Transformer (Vaswani et al., 2017). We posit that the router's computation is analogous to the first linear layer of a standard feed-forward network (FFN), which projects the input into a higher-dimensional space and can be interpreted as calculating a set of unnormalized scores or weights. The experts subsequently play the role of the second FFN layer, which operates on these weighted features to produce the final values (Geva et al., 2020). Inspired by structural similarities between MoE, Nadaraya-Watson regression, and FFN, we introduce a new class of simple yet effective router functions for MoE. Our primary proposed method defines an FFN-style router function, which generalizes both Softmax- and Sigmoid-based router functions. To align well with the practical and widely recognized FFN setups, we adopt the ReLU activation and a computationally lightweight $\ell_2$-normalization in the router function. This modification ensures that the magnitude of the MoE output is invariant with the number of experts, leading to more balanced expert participation and improved training stability without enforcing a probabilistic simplex constraint. Our key contributions are summarized as follows:

- **A Novel Perspective:** We reframe the MoE layer through the lens of the Nadaraya-Watson regression, interpreting it as a generalized FFN, providing a more flexible and principled view of expert aggregation.

- KERN **Router Function**: Motivated by the perspective of structure similarity, we propose Kernel Inspired Router with Normalization (KERN), a new family of simple yet effective FFN-style router functions. By introducing widely adopted ReLU activation and $\ell_2$-normalization, KERN promotes balanced expert utilization and stable training without the constraints or computational profile of Softmax, and crucially, without introducing any additional parameters or significant overhead.

- **Extensive Empirical Validation:** We conduct a comprehensive evaluation of KERN across a wide range of experimental setups, including varying model scales, sequence lengths, training dataset sizes and domains, and sparsity coefficients.

## 2 RELATED WORK

**Large Language Models.** With the inspiration of the language model scaling law (Kaplan et al., 2020), LLMs (Touvron et al., 2023; Achiam et al., 2023; Jiang et al., 2024; Liu et al., 2024a; Yang et al., 2025; Team et al., 2025; Comanici et al., 2025) have shown remarkable capabilities in a wide range of open-ended tasks, marking significant progress toward achieving general artificial intelligence. With the Transformer architecture (Vaswani et al., 2017), LLMs achieve significant performance in various areas, including reasoning (Achiam et al., 2023; Liu et al., 2024a; Team et al., 2025), language-visual model (Liu et al., 2023a; Jin et al., 2024a; Riquelme et al., 2021), language-audio (Yang et al., 2023; Rouditchenko et al., 2025) and so on.

**Mixture-of-Experts.** The MoE (Jacobs et al., 1991) is proposed to reduce the active parameters and aggregate the outputs of several models to reduce the training cost and empower expressiveness. With the development of LLMs, the MoE becomes increasingly attractive and dominant in applications of large-scale tasks (Achiam et al., 2023; Meta), where they must balance the load of experts (Lewis et al., 2021; Roller et al., 2021; Dai et al., 2024). The MoE originally presents its ability in the machine translation tasks (Shazeer et al., 2017). Later, Gshard (Lepikhin et al., 2021) proposes a more efficient implementation on parallel devices. To further improve the efficiency, Switch Transformer (Fedus et al., 2022) alternatively uses a single expert for one token prediction. Recently, a zero-cost expert (Jin et al., 2024b) is introduced, where the expert does not involve computation via skip connections. We notice that most of these works utilize Softmax as the router function.

**Feed-Forward Network.** FFN represents the standard neural network architecture, with origins tracing back to the early development of deep learning. Numerous studies have examined different activation functions and normalization techniques to enhance their expressiveness and training stability (Householder, 1941; Fukushima, 1980; 2007; Hendrycks & Gimpel, 2016). More recently, research has shifted toward understanding the role of FFNs within Transformer models, where they are often interpreted as a form of static memory, in contrast to the dynamic memory provided by attention mechanisms. From this perspective, Transformers can be viewed as integrating both static and dynamic memory, each contributing distinct modes of information processing (Liu et al., 2023b; Zhong et al., 2025).

## 3 METHOD

In this section, we first introduce the well-known Nadaraya–Watson regression and then compare its mathematical formulation with that of MoE. Motivated by their structural similarity, we reinterpret the MoE as a large FNN. Inspired by the new perspective of interpretation and the well-recognized FFN setups, we design an FFN-style router function of MoE, namely, KERN, which is equipped with ReLU activation and $\ell_2$-normalization. We also analyze the relationships and advantages of KERN, compared to widely recognized Softmax and Sigmoid router functions.

### 3.1 NADARAYA-WATSON REGRESSION

The Nadaraya-Watson estimator predicts the output $y$ for an input $x$ by assigning weights to training samples $\{(x_i, y_i)\}_{i=1}^N$ according to their similarity to $x$:

$$f_{\text{NW}}(x) = \sum_{i=1}^N \frac{K(x, x_i)}{\sum_{j=1}^N K(x, x_j)} y_i \tag{1}$$

where $K(\cdot, \cdot)$ is a kernel function measuring the similarity between two points. The most widely used choice is the Gaussian kernel formulated as $K(u, v) = \exp(-\|u - v\|^2 / 2\sigma^2)$, where the bandwidth $\sigma$ (standard deviation of the Gaussian distribution) controls the smoothness of the estimator.

In practice, the bandwidth $\sigma$ is typically unknown and treated as a hyperparameter. A more flexible approach is to regard $\sigma$ as a trainable parameter to be optimized. Equivalently, we define a parameterized kernel as $K(u, v; w) = \exp(-w\|u - v\|^2 / 2)$, where $w > 0$ is a learnable weight. The corresponding estimator becomes

$$f_{\text{NW}}(x; w) = \sum_{i=1}^N \frac{K(x, x_i; w)}{\sum_{j=1}^N K(x, x_j; w)} y_i. \tag{2}$$

This parametric formulation allows the model to adapt the kernel bandwidth during training, improving flexibility and performance in practice. Moreover, the idea naturally extends beyond Gaussian kernels that one can generalize to a learnable kernel of the form $K(u, v; w)$, parameterized by a vector $w$.

### 3.2 FFN AS PARAMETRIC NADARAYA-WATSON REGRESSION

The output layer of a standard FFN admits two potential formulations

$$\text{FFN}(x) = \sum_{i=1}^h \underbrace{\phi\left(\text{LN}\left(\langle w_i, \Phi(x) \rangle\right)\right)}_{\text{Adaptive kernel weight}} \cdot \underbrace{v_i}_{\text{Value}}, \tag{3}$$

$$\text{FFN}(x) = \sum_{i=1}^h \underbrace{\text{LN}\left(\phi\left(\langle w_i, \Phi(x) \rangle\right)\right)}_{\text{Adaptive kernel weight}} \cdot \underbrace{v_i}_{\text{Value}}, \tag{4}$$

where $\phi$ is the activation function in the FFN, $\Phi(x)$ denotes the hidden representation input to the output layer, and $V = [v_1, \ldots, v_h]$ are the output-layer weights. Here, $\text{LN}(\cdot)$ denotes layer normalization.

Comparing Equation (4) with the adaptive Nadaraya-Watson estimator in Equation (2), we see that the FFN implicitly defines a parameterized FFN-style kernel function

$$K(\boldsymbol{x}, \{\boldsymbol{w}_i, b_i\}) = \phi\left(\langle \boldsymbol{w}_i, \Phi(\boldsymbol{x}) \rangle\right), \tag{5}$$

where the normalization is applied after the kernel function, the role of the labels $\boldsymbol{y}_i$ is played by the value vectors $\boldsymbol{v}_i$, and $\Phi$ is a transformation function. In this analogy, the normalization step in Equation (2) corresponds to $\ell_1$-normalization $\mathrm{LN}(\boldsymbol{x}) = \frac{\boldsymbol{x}}{\|\boldsymbol{x}\|_1}$. This observation motivates a natural generalization of replacing the normalization $\mathrm{LN}(\cdot)$ in Nadaraya-Watson regression with a more commonly used $\ell_2$-normalization in FFN. This perspective provides a mathematical interpretation of the FFN as a special instantiation of parametric Nadaraya-Watson regression.

### 3.3 MIXTURE-OF-EXPERTS AS PARAMETRIC NADARAYA-WATSON REGRESSION

The MoE model combines multiple expert networks $\{E_m(\boldsymbol{x})\}_{m=1}^{M}$ through a router $\boldsymbol{g}(\boldsymbol{x})$:

$$\mathrm{MoE}(\boldsymbol{x}) = \sum_{m=1}^{M} g_m(\boldsymbol{x}) E_m(\boldsymbol{x}), \tag{6}$$

where the router $\boldsymbol{g}(\boldsymbol{x}) = [g_1(\boldsymbol{x}), \ldots, g_M(\boldsymbol{x})]$ admits Softmax outputs:

$$g_m(\boldsymbol{x}) = \frac{\exp(\langle \boldsymbol{w}_m, \boldsymbol{x} \rangle)}{\sum_{j=1}^{M} \exp(\langle \boldsymbol{w}_j, \boldsymbol{x} \rangle)}.$$

The structure in Equation (6) closely resembles Nadaraya-Watson regression in Equation (2) that the router weight $g_m(\boldsymbol{x})$ can be viewed as a kernel function $K(\boldsymbol{x}, \boldsymbol{w}_m)$, while each expert output $E_m(\boldsymbol{x})$ corresponds to an observation $\boldsymbol{y}_m$ being aggregated. Recall from the previous section that we generalized the kernel function to the FFN-style form given in Equation (5). Under this perspective, the MoE in Equation (6) can be interpreted and designed as a large network that aggregates expert networks $\{E_m(\boldsymbol{x})\}_{m=1}^{M}$ via such an FFN-style kernel function given by:

$$g_m(\boldsymbol{x}) = \phi\left(\mathrm{LN}\left(\langle \boldsymbol{w}_m, \Phi(\boldsymbol{x}) \rangle\right)\right). \tag{7}$$

### 3.4 KERN ROUTER FUNCTION

Let $\Phi(\boldsymbol{x}) \in \mathbb{R}^d$ denote the representation that feeds the router. We introduce a novel router function defined in Equation (7), namely, the kernel-inspired router with normalization (KERN), that instantiates the FFN-style router with a linear projection followed by (i) $\ell_2$-normalization, (ii) a ReLU activation, and (iii) an optimal learnable global scaler:

$$\begin{aligned} \boldsymbol{s}(\boldsymbol{x}) &= \boldsymbol{W}_s \Phi(\boldsymbol{x}) + \boldsymbol{b}_s, \\ \bar{\boldsymbol{s}}(\boldsymbol{x}) &= \frac{\boldsymbol{s}(\boldsymbol{x})}{\|\boldsymbol{s}(\boldsymbol{x})\|_2 + \varepsilon}, \\ \boldsymbol{r}(\boldsymbol{x}) &= \mathrm{ReLU}(\bar{\boldsymbol{s}}(\boldsymbol{x})), \\ \hat{\boldsymbol{g}}(\boldsymbol{x}) &= \gamma \cdot \boldsymbol{r}(\boldsymbol{x}), \end{aligned} \tag{8}$$

where $W_s \in \mathbb{R}^{M \times d}$ and $\boldsymbol{b}_s \in \mathbb{R}^M$ are the router parameters, $\varepsilon$ is a small constant that guards against division by zero, and $\gamma$ is a learnable scalar initialized to 1. The normalization step keeps the scale of the logits invariant to the number of experts $M$, while the ReLU activation preserves sparsity without resorting to exponential functions. **We further discussed the effect of ReLU in Appendix J.** During inference and training, we retain only the top-$k$ routed experts:

$$\mathcal{T}_k(\boldsymbol{x}) = \mathrm{TopKIndices}(\hat{\boldsymbol{g}}(\boldsymbol{x}), k), \tag{9}$$

$$g_m(\boldsymbol{x}) = \hat{g}_m(\boldsymbol{x}) \, \mathbf{1}[m \in \mathcal{T}_k(\boldsymbol{x})], \tag{10}$$

$$\mathrm{MoE}_{\mathrm{KERN}}(\boldsymbol{x}) = \sum_{m=1}^{M} g_m(\boldsymbol{x}) \, E_m(\boldsymbol{x}). \tag{11}$$

Because KERN does not project the router outputs onto the probability simplex, we do not perform an additional $\ell_1$ rescaling; the magnitude of $g_m(\boldsymbol{x})$ is instead controlled by the global scale $\gamma$ and the $\ell_2$ constraint. This simple construction matches the inductive biases of standard FFNs while avoiding the gradient saturation issues of exponential routers.

**Comparisons to existing router functions**. From this viewpoint, the standard MoE with a $\mathrm{Softmax}$ router corresponds to an FFN-style router where $\ell_1$-normalization is applied through $\mathrm{LN}(\cdot)$ and the exponential function serves as the activation. Interestingly, recent work has shown that replacing the $\mathrm{Softmax}$ with a $\mathrm{Sigmoid}$ router yields improved performance (Nguyen et al., 2024a; Chi et al., 2022; Wang et al., 2024b;a). It also admits an FFN interpretation that the router function reduces to a $\mathrm{Sigmoid}$ activation $\phi(\cdot)$ without layer normalization. Hence, our interpretation frames MoE routing as a general framework that encompasses the most widely adopted router functions. However, under this formulation, the commonly used $\mathrm{Softmax}$ and $\mathrm{Sigmoid}$ routers appear atypical when compared with standard FFN activations and layer normalization in deep learning. This observation motivates us to explore and design the router function KERN, which is more consistent with the classical FFN paradigm, as discussed in the next paragraph.

**Practical setups for the FFN-style router function** KERN. To better align with common practices in deep learning, we propose adopting router functions that mirror typical FFN designs, namely using a ReLU activation $\phi(\cdot)$ combined with widely-adopted $\ell_2$-normalization $\mathrm{LN}(\cdot)$. This choice is motivated by the following observations. First, exponential-type activations are rarely used in modern architectures, as they tend to be highly sensitive to input values, leading to rapid value explosion and gradient vanishing. In contrast, ReLU-type activations, or even linear outputs without nonlinearity, are far more common in practice, providing numerical stability and robustness during training. Second, although all vector norms are mathematically equivalent in finite-dimensional spaces, $\ell_2$-normalization remains the dominant choice in deep learning. Specifically, $\ell_2$-normalization stabilizes the variance of vectors, ensuring scale consistency and stable computation regardless of model size.

**Advantages of the** KERN **router function**. First, the gradient vanishing problem inherent in $\mathrm{Softmax}$ and $\mathrm{Sigmoid}$ (exponential-type) router functions can be alleviated by adopting the proposed FFN-style router (KERN) with appropriate activation functions. Prior studies have highlighted that $\mathrm{Softmax}$ and $\mathrm{Sigmoid}$ activations often suffer from saturation, that small values push experts toward near-inactivity, resulting in negligible gradient updates. Intuitively, if an expert stays at an almost-zero routing weight, the vanishing gradient problem can trap it in this poor state, preventing meaningful updates or improvements. In contrast, the proposed KERN reduces this risk. The gradients vanish less severely, ensuring that even less active experts still receive updates, therefore, promoting better expert utilization and training dynamics. Second, $\ell_2$-normalization in KERN preserves the variance of the MoE output at a constant scale. Since experts are independently and properly initialized, we may assume that the outputs of the $M$ experts, $\{E_m(\boldsymbol{x})\}_{m=1}^{M}$, are independent and have bounded norm (i.e., $\|E_m(\boldsymbol{x})\|_2 = \mathcal{O}(1)$). Under this assumption, the MoE with KERN at initialization satisfies

$$
\begin{aligned}
\mathbb{E}\left[\|\mathrm{MoE}_{\mathrm{KERN}}(\boldsymbol{x})\|_2^2\right] &= \mathbb{E}\left[\left\|\sum_{m=1}^{M} g_m(\boldsymbol{x})E_m(\boldsymbol{x})\right\|_2^2\right] \\
&= \sum_{m=1}^{M} (g_m(\boldsymbol{x}))^2\, \mathbb{E}\left[\|E_m(\boldsymbol{x})\|_2^2\right] \\
&= \mathcal{O}(1) \cdot \sum_{m=1}^{M} (g_m(\boldsymbol{x}))^2 = \mathcal{O}(1).
\end{aligned}
$$

The final equality holds for most commonly used activation functions $\phi$ (e.g., ReLU, LeakyReLU, Tanh, GeLU) when applied within the proposed FFN-style router function, where KERN adopts ReLU activation. This result demonstrates that $\ell_2$-normalization maintains the MoE output at a constant scale, thereby ensuring stable network computations and training. Such stability is consistent with the initialization principles commonly adopted in deep neural networks, e.g., Kaiming initialization.

## 4 EXPERIMENT

**Baseline.** We compare the proposed KERN with the Dense model and MoE with other router functions. To be specific, we evaluate KERN against a range of routers, including Softmax, Sigmoid, and Tanh. For the MoE and Dense model, they have the same active parameters. Additionally, for the MoE model, the ratio of active parameters and total parameters is 8, where there are 64 experts in total and 8 active experts. We utilize more than 8 active experts, as recent works propose using a larger number of active experts (Liu et al., 2024a; Team et al., 2025).

**Datasets.** Our analysis involves training language models on the Arxiv and Books3 datasets, which are frequently used benchmarks for evaluating model performance (Press et al., 2022). Moreover, we train the model on the large-scale dataset FinWeb-Edu (Penedo et al., 2024; Lozhkov et al., 2024) and evaluate on downstream datasets, including ARC (Clark et al., 2018), HellaSwag (Zellers et al., 2019), PIQA (Bisk et al., 2020), ScIQ (Welbl et al., 2017), and WinoGrande (Sakaguchi et al., 2021)

**Experiment settings.** Initially, we compare KERN with other baselines at training lengths 512 and 1024, using decoder-only Transformers (Brown et al., 2020) with model size 125M, whose configuration is shown in Appendix D. Subsequently, we evaluate the performance of larger model sizes, specifically 350M and 2.7 B. Finally, we analyze routers and MoE models by various active experts while holding the active ratio fixed, and we examine the effect of sparsity.

### 4.1 COMPARISONS WITH BASELINES

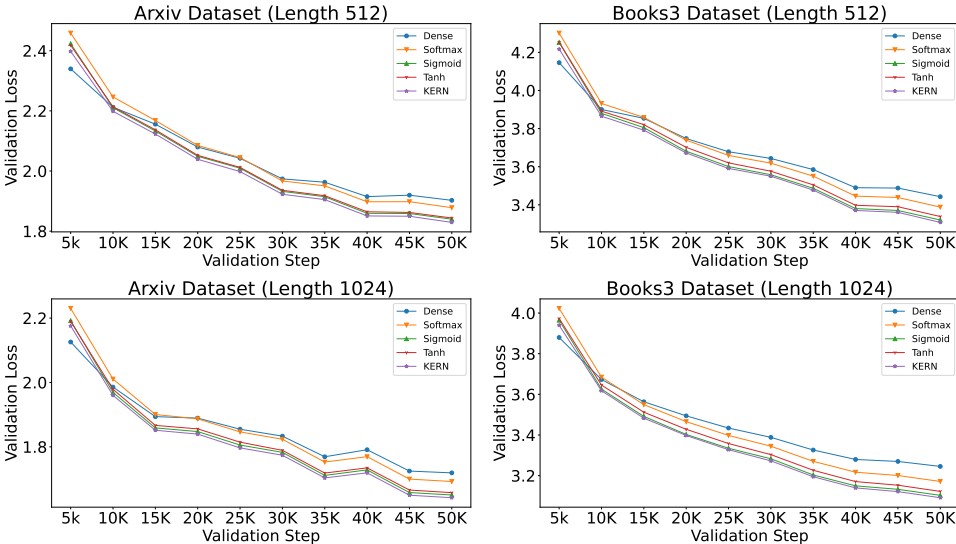

Figure 1: The performance of different methods on the Arxiv and Books3 dataset, with model parameter 520M, activated parameter 125M, training lengths of 512 and 1024.

KERN **achieves superior performance across different datasets.** We validate our method on the Arxiv and Books3 datasets in Figure 1. On Arxiv with training length 512, the Dense model (125M) reaches losses of 2.3396 and 1.0925 at steps 5,000 and 50,000, respectively. The Softmax router shows a higher initial loss (2.4586) but a better final loss (1.8781), suggesting a potentially slower convergence rate but strong final performance. Our KERN method achieves the best results at both checkpoints (2.3975 and 1.8291). A similar trend is observed on the Books3 dataset. The Dense model records 4.1460 (step 5,000) and 3.4429 (step 50,000). The Softmax router achieves 4.3011 and 3.3882. Once again, KERN delivers the best performance, with losses of 4.2165 and 3.3080 at steps 5,000 and 50,000, respectively. Therefore, regardless of the training dataset, KERN consistently achieves state-of-the-art performance.

KERN **achieves superior performance across varying training lengths.** We further evaluate model performances among various router functions using a context length of 512 on the Books3 dataset. The baseline Dense model achieves a loss of 3.4429. Among these MoE routers, Sigmoid (3.3206), Tanh (3.3388), and Softmax (3.3882) all outperform the Dense baseline. The proposed KERN method achieves the best performance with a loss of 3.3080. When the context length is increased to 1024, the performance ranking remains consistent: the Dense model attains a loss of 3.2454, while Softmax, Tanh, and Sigmoid achieve 3.1714, 3.1224, and 3.1031, respectively. Notably, KERN again achieves the lowest loss 3.0914, demonstrating its robustness across different training lengths.

KERN **achieves superior performance with longer context lengths.** The advantage of KERN is further demonstrated at a longer context length of 2048 in Figure 2. The baseline Dense model achieves a loss of 3.1249. The Softmax router shows an improvement with a loss of 3.0442, while Sigmoid (2.9635) and Tanh (2.9868) perform better still. The proposed KERN method achieves the best performance overall, with a lowest loss of 2.9535. These results confirm that KERN maintains its effectiveness and superiority as the training length increases.

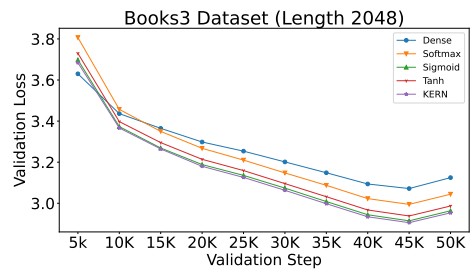

Figure 2: The performance on training length 2048.

## 4.2 The Performance on Large Language Models

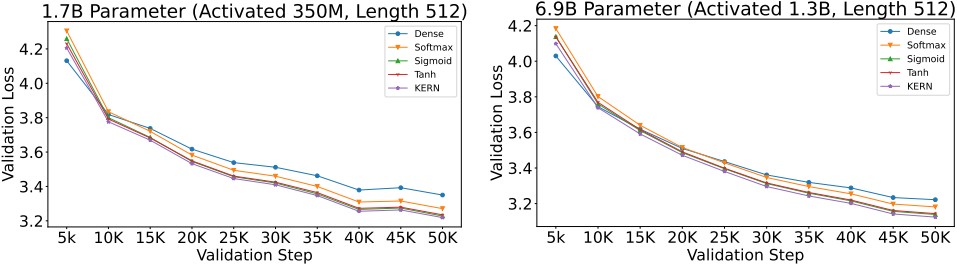

Figure 3: The performance of different methods on the Books3 dataset, with active model size 350M and 1.3B.

**The performance gap provided by** KERN **is maintained for large models.** For Figure 8, while increasing the active model size from 125M to 1.3B parameters reduces the loss for all baseline methods, KERN consistently achieves the best performance at every scale. At 350M parameters, its loss of 3.2188 is lower than that of the comparable Softmax (3.2709) and Dense (3.3500) models. This lead is extended at the 1.3B scale, where KERN's loss of 3.1241 significantly outperforms the Softmax (3.1814) and Dense (3.2219) results. This demonstrates that KERN is not only effective but is particularly advantageous for training larger-scale models.

## 4.3 The Effect of Granularity

KERN **consistently outperforms** Softmax **across expert counts.** Figure 4 provides compelling evidence for the practical superiority of KERN. When evaluated against the standard Softmax baseline, our method achieves higher performance regardless of the number of experts activated during inference—a parameter we varied from 4 to 32. This demonstrates the robustness of our method irrespective of the specific capacity used during inference.

KERN **achieves better performance than other routers with small granularity (e.g., more experts, smaller expert size).** As shown in Figure 7, with 256 experts, 8 active experts, and an expert intermediate size is 96, the Softmax achieves 4.3139 loss at evaluation step 5K and 3.4150 loss

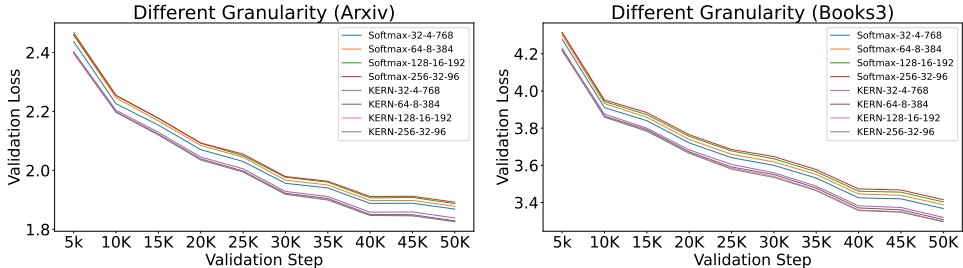

Figure 4: The performance of different methods on the Arxiv and Books3 dataset, with different active expert numbers and the same active parameter number. **The** $\mathrm{Softmax} - 64 - 8 - 384$ **suggests that the router function is** $\mathrm{Softmax}$**, and there are 64 experts, 8 active experts, and each expert's intermediate size is 384.**

at evaluation step 50K. The $\mathrm{Sigmoid}$ and $\mathrm{Tanh}$ achieve 4.2924 loss and 4.2435 loss at the evaluation step 5K, and $\mathrm{Sigmoid}$ and $\mathrm{Tanh}$ achieve 3.3302 loss and 3.3276 loss at the evaluation step 50K. The $\mathrm{KERN}$ achieves the best performance 4.2294 loss at evaluation step 5K and 3.2962 loss at evaluation step 50K. Therefore, $\mathrm{KERN}$ achieves better performance than other routers with small granularity.

## 4.4 THE EFFECT OF SPARSITY

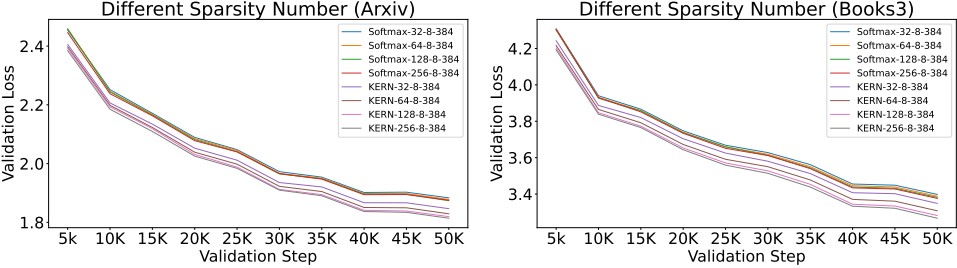

Figure 5: The performance of different methods on the Arxiv and Books3 dataset, with different total expert numbers and the same active parameter number. **The** $\mathrm{Softmax} - 64 - 8 - 384$ **suggests that the router function is** $\mathrm{Softmax}$**, and there are 64 experts, 8 active experts, and each expert's intermediate size is 384.**

$\mathrm{KERN}$ **outperforms** $\mathrm{Softmax}$ **across all sparsity levels.** As evidenced by Figure 5, $\mathrm{KERN}$ achieves a lower loss than the $\mathrm{Softmax}$ baseline for every total number of experts tested (32 to 256) on both the Books3 and Arxiv datasets. On Books3, $\mathrm{KERN}$'s loss (3.3487, 3.3080, 3.2820, 3.2672) is consistently superior to $\mathrm{Softmax}$'s (3.3981, 3.3882, 3.3817, 3.3761). This trend holds on the Arxiv dataset, where $\mathrm{KERN}$'s results (1.8466, 1.8291, 1.8195, 1.8141) are consistently better than those of $\mathrm{Softmax}$ (1.8835, 1.8781, 1.8738, 1.8754). This demonstrates that the performance gain of $\mathrm{KERN}$ is robust to changes in model sparsity.

$\mathrm{KERN}$ **achieves better performance than other routers with large sparsity.** As shown in Figure 6, with 256 experts, 8 active experts, and an expert intermediate size of 384, the $\mathrm{Softmax}$ achieves 4.3059 loss at evaluation step 5K and 3.3761 loss at evaluation step 50K. The $\mathrm{Sigmoid}$ and $\mathrm{Tanh}$ achieve 4.2924 loss and 4.2435 loss at the evaluation step 5K, and $\mathrm{Sigmoid}$ and $\mathrm{Tanh}$ achieve 3.2760 loss and 3.2972 loss at the evaluation step 50K. The $\mathrm{KERN}$ achieves the best performance 4.1926 loss

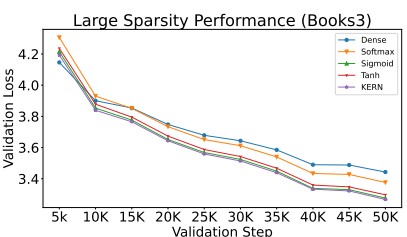

Figure 6: The performance with 256 experts, 8 active experts and each expert's intermediate dimension is 384.

at evaluation step 5K and 3.2672 loss at evaluation step 50K. Therefore, KERN achieves better performance than other routers with large sparsity.

## 4.5 THE PERFORMANCE ON LARGE-SCALE PRETRAIN DATASET

Table 1: Main language modeling results against different methods. All models are trained on the same subset of the FineWeb-Edu dataset (Penedo et al., 2024; Lozhkov et al., 2024) with the GPT-2 tokenizer.

| Model | ARC-E | ARC-C | Hellaswag | PIQA | ScIQ | Winograde | Avg |
|---|---|---|---|---|---|---|---|
| *520M (Active 125M) params* | | | | | | | |
| Dense | 47.35 | 20.48 | 31.59 | 66.21 | 73.60 | 51.85 | 48.51 |
| Softmax | 49.49 | 20.39 | 34.81 | 69.42 | 75.50 | 49.64 | 49.88 |
| Tanh | 51.60 | 21.59 | 35.90 | 69.91 | 76.90 | 53.28 | 51.53 |
| Sigmoid | 51.89 | 20.99 | 37.11 | 70.78 | **78.50** | 51.54 | 51.80 |
| KERN | **53.32** | **21.67** | **37.12** | **70.89** | 77.80 | **52.01** | **52.14** |
| *1.7B (Active 350M) params* | | | | | | | |
| Dense | 51.26 | 22.10 | 35.07 | 70.13 | 77.60 | 50.12 | 51.05 |
| Softmax | 51.94 | 22.78 | 37.65 | 70.13 | 79.70 | 52.57 | 52.46 |
| Tanh | 55.51 | 23.21 | 40.30 | 72.03 | 81.60 | 52.41 | 54.18 |
| Sigmoid | **56.61** | 23.81 | **40.78** | 72.74 | 80.40 | **53.99** | 54.72 |
| KERN | 56.48 | **24.40** | 40.68 | **73.61** | **82.00** | 53.59 | **55.13** |
| *6.9B (Active 1.3B) params* | | | | | | | |
| Dense | 58.59 | 24.15 | 42.36 | 72.85 | 82.90 | 55.80 | 56.11 |
| Softmax | 59.51 | 23.55 | 42.29 | 73.18 | 84.70 | 55.72 | 56.49 |
| Tanh | 61.20 | 26.79 | 45.01 | 73.29 | 85.20 | 56.75 | 58.04 |
| Sigmoid | **62.33** | 27.47 | 45.52 | 74.43 | 84.70 | 56.83 | 58.55 |
| KERN | 61.20 | **27.90** | **45.95** | **75.19** | **84.90** | **58.17** | **58.88** |

**Downstream Evaluation.** We evaluate performance on standard benchmarks, including ARC (Clark et al., 2018), HellaSwag (Zellers et al., 2019), PIQA (Bisk et al., 2020), ScIQ (Welbl et al., 2017), and WinoGrade (Sakaguchi et al., 2021), using the `lm-evaluation-harness` (Gao et al., 2024) codebase. The evaluation metric is the accuracy. We train the model with 50K steps with training length 1024 and training tokens 50B. The model sizes are 520M (active 125M), 1.7B (active 350M), and 6.9B (active 1.3B). We display the zero-shot evaluation results of models here in Tables 1.

**With the same active parameter, the KERN is always better than the routers, from small model size (e.g., 125M active) to larger model size (e.g., 1.3B active).** At a 520M model size (125M active), KERN achieves an average performance of 52.14, surpassing Dense (48.51), Softmax (49.88), Tanh (51.53), and Sigmoid (51.80). With a 1.7B model size (350M active), it scores 55.13, outperforming Dense (51.05), Softmax (52.46), Tanh (54.18), and Sigmoid (54.72). Similarly, at 6.9B (1.3B active), it reaches 58.88, exceeding Dense (56.11), Softmax (56.49), Tanh (58.04), and Sigmoid (58.55). Therefore, KERN is always better than the routers, from small model size to larger model size.

**With the same active parameters, the performance gap between KERN and Softmax is comparable to that between Softmax and Dense model.** For a 520M model (125M active), the performance gap between KERN and Softmax is 2.26, compared to the 1.37 gap between Softmax and Dense. With a 1.7B model (350M active), the KERN-Softmax gap widens to 2.67, while the Softmax-Dense gap is 0.61. At the 6.9B scale (1.3B active), the KERN-Softmax gap remains significant at 2.39, vastly exceeding the 0.38 gap between Softmax and Dense. The MoE model achieves a significant performance gain over the Dense model. Since KERN achieves an even larger gain at zero additional cost, it should be a critical component for model sparsity.

## 5 CONCLUSION

In general, the use of the Softmax function has been the de facto standard for generating router scores in MoE models. In this work, we challenge this convention by recasting MoE routing through the novel lens of the Nadaraya-Watson estimator. Motivated by this perspective, we introduce the

novel KERN router function for MoE, an FFN-style kernel function with ReLU activation and $\ell_2$-normalization. We extensively validate the efficacy of these functions through comprehensive experiments across varying model scales, sequence lengths, and, most significantly, in large-scale pre-training followed by downstream task evaluation. Our empirical results demonstrate that these simpler alternatives are not only viable but often match or exceed the performance of Softmax-based routing. We believe this work opens a new direction for router design and anticipate that KERN will serve as a strong baseline and a potential substitute for Softmax in future MoE architectures.

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

## A  ETHICS STATEMENT

This research work is fundamentally focused on the architectural and algorithmic enhancement of the Mixture-of-Experts (MoE) model paradigm. Our primary contribution involves a novel integration of non-parametric kernel regression methods, specifically the Nadaraya-Watson estimator, to re-formulate the gating mechanism traditionally governed by the $\mathrm{Softmax}$ function. This approach replaces the standard $\mathrm{Softmax}$-based probability distribution with a kernel-smoothed weighting scheme based on the $\mathrm{KERN}$ between an input token and each expert's representative vector. Consequently, this research does not introduce any novel, domain-specific ethical claims or societal impacts that diverge from the well-documented considerations already associated with large-scale language models in general.

## B  REPRODUCIBILITY STATEMENT

A comprehensive elucidation of the proposed methodology is presented in Section 3, which details the theoretical foundations and algorithmic structure of our approach. To ensure reproducibility and facilitate further research, we have made our complete code implementation publicly available. This code, which includes scripts for training, inference, and analysis, is comprehensively documented in Appendix L. Furthermore, the complete set of hyperparameters, architectural details, and training configurations for all models discussed in our experiments are provided in Appendix D.

## C  THE USE OF LLMS

For this work, we mainly use the Large Language Model to aid or polish writing.

## D  MODEL CONFIGURATION

All experiments are conducted on 2 GPUs. The 125M and 350M model configuration is the following.

Table 2: **Model Configurations.**

|  | **125M** | **350M** |
| --- | --- | --- |
| Training sequence length | 512 | 512 |
| Batch size | $16 \times 2$ | $16 \times 2$ |
| Number of iterations | 50k | 50k |
| Dropout prob. | 0.0 | 0.0 |
| Attention dropout prob. | 0.0 | 0.0 |
| Attention head | 12 | 16 |
| Feature dimension | 768 | 1024 |
| Layer number | 12 | 24 |
| Optimizer | Adam | Adam |
| Optimizer parameter betas | [0.9, 0.95] | [0.9, 0.95] |
| Learning rate | $6e-4$ | $3e-4$ |
| Precision | float16 | float16 |
| Total Expert Number | 64 | 64 |
| Active Expert Number | 8 | 8 |

## E  TIME COST AND COMPUTATIONAL COST

**Theoretically, the proposed method does not have additional cost.**  A central advantage of the proposed gating mechanism is its computational parsimony. The core operation involves calculating the L2-norm for each expert's representation vector and for the input token's projection. The primary operation—division by the L2-norm—constitutes an element-wise operation. Therefore, when

analyzed from a theoretical perspective, the proposed router introduces no substantive additional time cost compared to the conventional Softmax-based approach, making it an efficient drop-in replacement.

# F  PERFORMANCE WITH SMALL GRANULARITY

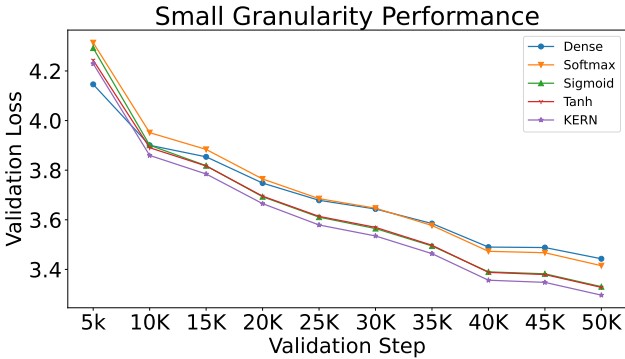

Figure 7: The performance with 256 experts, 32 active experts and each expert's intermediate dimension is 96.

# G  THE TRAINING LOSS WITH DIFFERENT METHODS

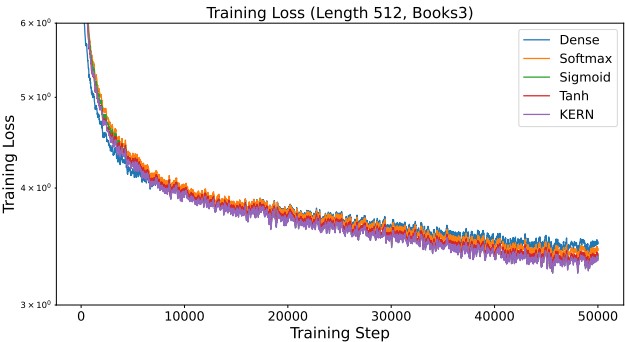

Figure 8: The performance of different methods on the Books3 dataset, with active model size 125M and training length 512. There are 64 experts and 8 active experts.

The Dense model demonstrates a characteristically rapid initial learning phase, achieving a swift and substantial reduction in loss during the early stages of training. This aggressive early convergence suggests a highly efficient optimization landscape for simpler, parameter-dense architectures, allowing them to quickly capitalize on low-hanging fruit within the dataset. However, this initial advantage is not sustained over the long term. As the number of training steps increases, the Dense model's loss curve begins to exhibit signs of stagnation and ultimately plateaus at a higher value than its MoE counterparts. This pattern indicates that while the Dense model is easier to converge to a reasonable solution, it is ultimately constrained by its architectural limitations. The monolithic nature of its parameters appears to create a lower performance ceiling, limiting its capacity to capture the complex, nuanced patterns present in the data. In essence, it finds a good solution quickly but lacks the expressive power to find a great one. In stark contrast, the MoE model, particularly the one enhanced with the KERN technique, exhibits a profoundly different and more powerful learning trajectory. While its initial loss reduction may be marginally less explosive than the Dense model's, it demonstrates remarkable consistency and resilience throughout the entire training process. The KERN model does not merely converge; it continues to refine its performance, driving the loss to a

significantly lower plateau. This sustained improvement underscores a superior capacity for learning and generalization

## H  THE IMPORTANCE OF CONSIDERING ALL ROUTER LOGIT

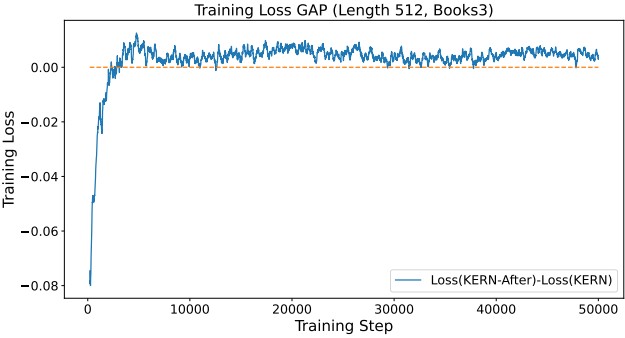

Figure 9: The performance of different methods on the Books3 dataset, with active model size 125M and training length 512. KERN: use KERN before Top-K choice. KERN-After: use KERN after Top-K choice. The result is the Loss(KERN-After)-Loss(KERN).

According to the caption in Figure 9, the result shown is the difference in loss (Loss(KERN-After) - Loss(KERN)). Therefore, a positive value indicates that KERN has a lower loss and thus performs better. Initially, the loss difference is negative, meaning KERN-After has a lower loss and performs better in early training. As training progresses, the difference becomes positive, indicating that KERN gradually achieves superior performance. This suggests that considering all router logits is important for better final performance.

## I  THE EFFECT OF KERN INITIALIZATION

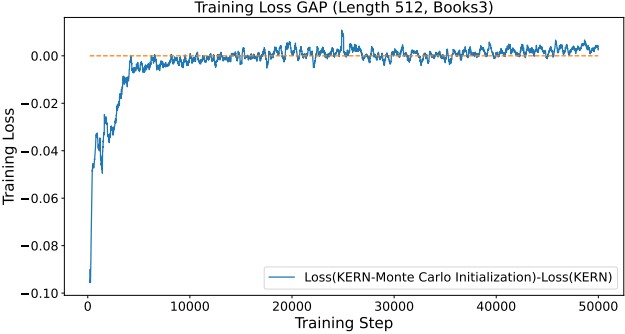

Figure 10: The performance of different methods on the Books3 dataset, with an active model size of 125M and training length of 512. KERN: the initial scale is 1. KERN-Monte Carlo Initialization: the Monte Carlo for the initialization, presented in Appendix L.

According to the caption in Figure 10, the result shown is the difference in loss (Loss(KERN-Monte Carlo Initialization) - Loss(KERN)). Therefore, a positive value indicates that KERN has a lower loss and thus performs better. Initially, the loss difference is negative, meaning KERN-Monte Carlo Initialization has a lower loss and performs better in early training. As training progresses, the difference becomes close to zero and slightly positive, indicating that KERN gradually achieves superior performance.

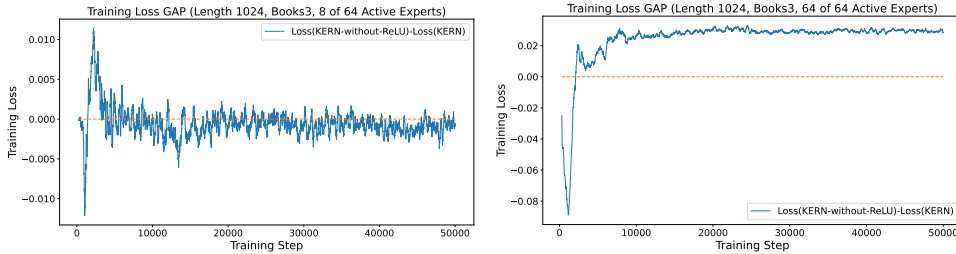

Figure 11: The performance of different methods on the Books3 dataset, with model parameter 520M.

## J  THE EFFECT OF RELU IN KERN

We compare the performance between KERN-without-ReLU and KERN in Figure 11. When the active expert number is less than half of the total number, the KERN achieves comparable performance with the KERN-without-ReLU, as the select router weights are usually positive. However, when the active expert number is higher (such as active all experts), the KERN will achieve significantly better performance than KERN-without-ReLU.

Table 3: The validation loss of different methods, with training length 1024 and Books3 dataset.

| Model | 5K | 10K | 15K | 20K | 25K | 30K | 35K | 40K | 45K | 50K |
|---|---|---|---|---|---|---|---|---|---|---|
| 8 of 64 Active Expert | | | | | | | | | | |
| Softmax | 4.0230 | 3.6848 | 3.5494 | 3.4658 | 3.3979 | 3.3447 | 3.2704 | 3.2168 | 3.2007 | 3.1714 |
| Sigmoid | 3.9648 | 3.6256 | 3.4915 | 3.4027 | 3.3347 | 3.2830 | 3.2033 | 3.1498 | 3.1327 | 3.1031 |
| Tanh | 3.9729 | 3.6457 | 3.5121 | 3.4281 | 3.3580 | 3.3034 | 3.2264 | 3.1708 | 3.1529 | 3.1224 |
| KERN-without-ReLU | 3.9398 | 3.6178 | 3.4827 | 3.3975 | 3.3278 | 3.2727 | 3.1946 | 3.1397 | 3.1214 | 3.0914 |
| KERN | 3.9391 | 3.6180 | 3.4842 | 3.3959 | 3.3281 | 3.2725 | 3.1954 | 3.1396 | 3.1227 | 3.0925 |
| 64 of 64 Active Expert | | | | | | | | | | |
| Softmax | 3.9875 | 3.6551 | 3.5127 | 3.4252 | 3.3539 | 3.2972 | 3.2191 | 3.1641 | 3.1470 | 3.1161 |
| Sigmoid | 3.9944 | 3.6420 | 3.4991 | 3.4085 | 3.3363 | 3.2818 | 3.2035 | 3.1480 | 3.1299 | 3.1003 |
| Tanh | 4.0086 | 3.6694 | 3.5321 | 3.4489 | 3.3801 | 3.3280 | 3.2491 | 3.1910 | 3.1730 | 3.1408 |
| KERN-without-ReLU | 3.9695 | 3.6467 | 3.5045 | 3.4200 | 3.3508 | 3.2945 | 3.2139 | 3.1579 | 3.1385 | 3.1075 |
| KERN | 3.9513 | 3.6170 | 3.4788 | 3.3885 | 3.3199 | 3.2652 | 3.1861 | 3.1290 | 3.1081 | **3.0780** |

When the active expert number is the same as the total expert, the KERN achieves better performance than KERN-without-ReLU and all other methods. We compare the performance between KERN-without-ReLU, KERN, and other methods with all experts being active in Table 3. The KERN achieves the best performance when the expert activation ratio is relatively high, such as 64 of 64 active experts.

## K  PERFORMANCE WITH DIFFERENT SEEDS

Table 4: The validation loss with three random seeds, with training length 512 and Books3 dataset

| Model | | 5K | 10K | 15K | 20K | 25K | 30K | 35K | 40K | 45K | 50K |
|---|---|---|---|---|---|---|---|---|---|---|---|
| Dense | Mean | 4.1536 | 3.9179 | 3.8427 | 3.7583 | 3.6899 | 3.6341 | 3.5589 | 3.5024 | 3.4737 | 3.4458 |
| | Variance | 0.0064 | 0.0193 | 0.0078 | 0.0148 | 0.0097 | 0.0066 | 0.0299 | 0.0227 | 0.0117 | 0.0068 |
| Softmax | Mean | 4.3143 | 3.9502 | 3.8470 | 3.7472 | 3.6687 | 3.6068 | 3.5228 | 3.4569 | 3.4197 | 3.3873 |
| | Variance | 0.0094 | 0.0196 | 0.0083 | 0.0136 | 0.0106 | 0.0086 | 0.0290 | 0.0213 | 0.0140 | 0.0050 |
| Sigmoid | Mean | 4.2605 | 3.9019 | 3.7964 | 3.6945 | 3.6153 | 3.5507 | 3.4645 | 3.3957 | 3.3571 | 3.3236 |
| | Variance | 0.0060 | 0.0210 | 0.0078 | 0.0149 | 0.0131 | 0.0053 | 0.0277 | 0.0222 | 0.0108 | 0.0041 |
| Tanh | Mean | 4.2596 | 3.9078 | 3.8077 | 3.7063 | 3.6267 | 3.5615 | 3.4751 | 3.4058 | 3.3682 | 3.3351 |
| | Variance | 0.0049 | 0.0173 | 0.0099 | 0.0105 | 0.0094 | 0.0111 | 0.0302 | 0.0204 | 0.0163 | 0.0050 |
| KERN | Mean | 4.2281 | 3.8825 | 3.7819 | 3.6823 | 3.6031 | 3.5385 | 3.4520 | 3.3828 | 3.3446 | 3.3112 |
| | Variance | 0.0082 | 0.0194 | 0.0079 | 0.0131 | 0.0120 | 0.0085 | 0.0282 | 0.0211 | 0.0120 | 0.0041 |

Table 7 presents the performance comparison of five different model architectures (Dense, Softmax, Sigmoid, Tanh, and KERN) across various training steps (from 5K to 50K) using three different random seeds. The analysis reveals several key observations: All models demonstrate a consistent pattern of performance improvement as training progresses from 5K to 50K steps. The loss values decrease monotonically for all architectures, indicating successful learning convergence. The KERN model consistently achieves the best performance across all training milestones, followed closely by Sigmoid and Tanh architectures. The Dense model exhibits significantly higher variance compared to other architectures, particularly in the early training stages (0.1699 variance at 5K steps). This suggests that the Dense architecture is more sensitive to random seed initialization. In contrast, the specialized activation functions (Softmax, Sigmoid, Tanh, and KERN) show much lower variance, indicating more stable and consistent performance across different random seeds. At the final training stage (50K steps), the KERN model achieves the best performance with a mean loss of 3.3112, followed by Sigmoid (3.3236) and Tanh (3.3351). The Dense model performs the worst with a mean loss of 3.6030, indicating that specialized activation functions provide substantial performance benefits for this task. All models show the most rapid improvement during the initial training phases (5K-20K steps), with the rate of improvement gradually slowing in later stages. This pattern suggests that while additional training continues to provide benefits, the marginal gains diminish as the models approach their performance limits on this dataset. The results demonstrate that careful selection of activation functions and normalization techniques can significantly impact model stability and final performance, with the KERN architecture emerging as the most robust and effective choice for this particular task and dataset configuration.

## L    IMPLEMENTATION DETAILS

In this section, we present the implementation of the proposed KERN module in `PyTorch`.

```python
from tqdm import tqdm
import numpy as np

import torch
import torch.nn as nn

def relu(x):
    """
    Implements the Rectified Linear Unit (ReLU) activation function.

    Args:
      x: A NumPy array or a single numerical value.

    Returns:
      A NumPy array or a single numerical value with negative values replaced by zero.
    """
    return np.maximum(0, x)

def monte_carlo_y_k(d, k, num_samples=100000):
    samples = []

    for _ in tqdm(range(num_samples)):
        x = np.random.randn(d)
        y = x / np.linalg.norm(x)
        y = relu(y)
        y_sorted = np.sort(y)[::-1]
        y_k = y_sorted[:k]
        samples.append(1 / (y_k**2).sum() ** 0.5)

    return np.mean(samples)

class KERNRouter(nn.Module):
    def __init__(
        self,
        initial_method="one",
        total_expert=64,
        top_k=8,
        eps=1e-8,
    ):
        super().__init__()

        if initial_method == "one":
            self.scale_initial = 1
```

```
        elif initial_method == "monte_carlo":
            self.scale_initial = monte_carlo_y_k(total_expert, top_k)
        self.scale = nn.Parameter(torch.ones(1))
        self.eps = eps
        self.activation = nn.ReLU()

    def forward(self, x):
        norm_x = x.norm(2, dim=-1, keepdim=True)
        x_normed = x / (norm_x + self.eps)
        x_normed=self.activation(x_normed)

        return self.scale * self.scale_initial * x_normed
```

## M    PERFORMANCE WITH SINKHORN

Table 5: The train loss with sinkhorn, with training length 512 and Books3 dataset

| Model | 5K | 10K | 15K | 20K | 25K | 30K | 35K | 40K | 45K | 50K |
|---|---|---|---|---|---|---|---|---|---|---|
| Dense | 4.1460 | 3.9004 | 3.8538 | 3.7480 | 3.6786 | 3.6435 | 3.5850 | 3.4900 | 3.4881 | 3.4429 |
| Softmax | 4.3198 | 3.9538 | 3.8208 | 3.7427 | 3.6518 | 3.5777 | 3.5167 | 3.4380 | 3.3966 | 3.3607 |
| Sigmoid | 4.2990 | 3.9387 | 3.8034 | 3.7263 | 3.6374 | 3.5640 | 3.5034 | 3.4233 | 3.3804 | 3.3469 |
| KERN | 4.2693 | 3.9241 | 3.7910 | 3.7127 | 3.6266 | 3.5525 | 3.4928 | 3.4130 | 3.3694 | 3.3345 |

## N    EXPERT LOAD BALANCE

We calculate the KL loss for the expert load frequency and uniform distribution. The KERN achieves the lowest KL loss, suggesting that its expert load is more balanced. And the variance is also smaller.

Table 6: The KL loss for expert load frequency and uniform distribution. The KL is smaller, the load is more balanced.

| Method | Softmax | Sigmoid | KERN |
|---|---|---|---|
| KL Mean | 0.9483 | 0.3162 | 0.2947 |
| KL Standard Variance | 0.1997 | 0.2207 | 0.1896 |

## O    THE ODER OF NORM AND ACTIVATION FUNCTION

Table 7: The train loss with sinkhorn, with training length 512 and Books3 dataset

| Model | 5K | 10K | 15K | 20K | 25K | 30K | 35K | 40K | 45K | 50K |
|---|---|---|---|---|---|---|---|---|---|---|
| KERN (Equation 3) | 4.2228 | 3.8683 | 3.7954 | 3.6800 | 3.5978 | 3.5527 | 3.4795 | 3.3729 | 3.3647 | 3.3132 |
| KERN (Equation 4) | 4.2165 | 3.8650 | 3.7919 | 3.6724 | 3.5910 | 3.5503 | 3.4776 | 3.3707 | 3.3608 | 3.3080 |

