# OpenReview forum: "Understanding the Mixture-of-Experts with Nadaraya-Watson Kernel"
_ICLR.cc/2026/Conference — ICLR 2026 Poster_

### Official Review · Reviewer_XdAq · 2025-10-28

**Soundness:** 3
**Presentation:** 3
**Contribution:** 2
**Rating:** 6
**Confidence:** 3

**Summary:**

This paper revisits the design of the Mixture-of-Experts (MoE) routing mechanism through the lens of Nadaraya–Watson regression, a classical kernel-based estimator. The authors identify a structural similarity between MoE routers, feedforward networks (FFNs), and kernel smoothers, and use this observation to reinterpret the MoE routing process as a form of parametric kernel regression. Building on this perspective, the authors propose a new router function called KERN (Kernel Inspired Router with Normalization), which replaces the traditional Softmax gating with a lightweight $l_2$-normalization and ReLU activation. KERN eliminates the need for projecting gating scores onto a probability simplex, simplifying training and improving numerical stability. Empirical evaluations show that KERN achieves balanced expert utilization and improved stability in large-scale MoE and LLM setups, without adding computational or parameter overhead. The approach generalizes both Softmax- and Sigmoid-based routers while maintaining the inductive biases of standard FFN layers.

**Strengths:**

1. The paper offers a principled reinterpretation of Mixture-of-Experts (MoE) routing through the lens of Nadaraya–Watson regression, providing theoretical clarity and unifying MoE with FFN and kernel regression models.
2. The paper is well-organized, progressing clearly from Nadaraya–Watson regression to FFN interpretation, MoE formulation, and the final KERN methodology.
3. The authors conduct sufficient experiments and ablation studies on multiple MoE setups, validating both the theoretical claims and practical benefits of KERN.

**Weaknesses:**

1. The paper appears to change the order of operations between normalization and activation layers. Specifically, Equation (4) applies the activation function before normalization, which is inconsistent with Equation (3), where normalization precedes activation. This discrepancy may affect the theoretical consistency of the proposed analogy between FFN and Nadaraya–Watson regression.
2. The main contribution focuses on designing an alternative router function for MoE. While the proposed KERN router is elegant, its generality appears limited compared to broader architectural components like the Softmax function, which is widely used beyond MoE.
3. The paper would benefit from a figure that clearly presents the computational flow of the proposed router function. Although the mathematical formulation is clear, a visual depiction would enhance understanding and accessibility, especially for readers less familiar with MoE internals.

**Questions:**

1. How does changing the order between normalization and activation (as seen in Eq. (4) vs. Eq. (3)) influence the theoretical interpretation or empirical behavior of the model? Has this difference been empirically tested or justified in the paper?
2. Can the proposed KERN router be extended beyond MoE architectures to serve as a general routing or attention mechanism in neural networks, similar to how Softmax is used broadly across architectures?

---

> ### Author Response · Authors · 2025-11-14
> **Response to Reviewer XdAq (Part 1/2)**
>
> Dear Reviewer XdAq,
>
>  Thank you for the support. We address the concerns below.
>
>  **Q1: The paper appears to change the order of operations between normalization and activation layers. Specifically, Equation (4) applies the activation function before normalization, which is inconsistent with Equation (3), where normalization precedes activation.**
>
>  A1: **The normalization is before the activation function for KERN implementation. The FFN actually has two formulations, depending on the position of normalization:**
> * Formulation 1  (Normalization is before $phi$)):$FFN(x) = \sum_{i=1}^{h} \phi(LN(\langle w_i, \Phi(x)\rangle))v_i$, presented in previous work [1].
> * Formulation 2 (Normalization is after $phi$):$FFN(x) = \sum_{i=1}^{h} LN(\phi((\langle w_i, \Phi(x)\rangle))v_i$,  presented in previous work [2].
>
> **We find that formulation 1 is more widely used and actually works better for KERN, so that Equation 3 places the norm before $\phi$**
>
> .
> **We have presented the implementation in Appendix L Implementation Details**.
>
>
> ```python
> class KERNRouter(nn.Module):
>     def __init__(
>         self,
>         initial_method="one",
>         total_expert=64,
>         top_k=8,
>         eps=1e-8,
>     ):
>         super().__init__()
>
>         if initial_method == "one":
>             self.scale_initial = 1
>         elif initial_method == "monte_carlo":
>             self.scale_initial = monte_carlo_y_k(total_expert, top_k)
>         self.scale = nn.Parameter(torch.ones(1))
>         self.eps = eps
>         self.activation = nn.ReLU()
>
>     def forward(self, x):
>         norm_x = x.norm(2, dim=-1, keepdim=True)
>         x_normed = x / (norm_x + self.eps)
>         x_normed=self.activation(x_normed)
>
>         return self.scale * self.scale_initial * x_normed
> ```
>
>
>
> **Q2: The main contribution focuses on designing an alternative router function for MoE. While the proposed KERN router is elegant, its generality appears limited compared to broader architectural components like the Softmax function, which is widely used beyond MoE**
>
> A2: **The KERN could also be used in FFN and Attention Mechanism.**
>
> |MoE Router | Attention Mechnism | 5K | 10K | 15K | 20K | 25K | 30K | 35K | 40K | 45K | 50K |
> |---|---|----|-----|-----|-----|-----|-----|-----|-----|-----|-----|
> | Softmax  | Softmax  | 4.3011 | 3.9318 | 3.8586 | 3.7394 | 3.6594 | 3.6185 | 3.5507 | 3.4459 | 3.4387 | 3.3882
> | KERN  | KERN |      4.3307 | 3.9108 | 3.8262 | 3.7052 | 3.6229 | 3.5790 | 3.5070 | 3.3998 | 3.3918 | 3.3422
> | KERN  | Softmax  | **4.2165** | **3.8650** | **3.7919** | **3.6724** | **3.5910** | **3.5503** | **3.4776** | **3.3707** | **3.3608** | **3.3080**
>
>
> **Q3: The paper would benefit from a figure that clearly presents the computational flow of the proposed router function. Although the mathematical formulation is clear, a visual depiction would enhance understanding and accessibility, especially for readers less familiar with MoE internals.**
>
> A3: **We have presented the implementation in Appendix L Implementation Details**
>
> ```python
> from tqdm import tqdm
> import numpy as np
>
> import torch
> import torch.nn as nn
>
> def relu(x):
>   """
>   Implements the Rectified Linear Unit (ReLU) activation function.
>
>   Args:
>     x: A NumPy array or a single numerical value.
>
>   Returns:
>     A NumPy array or a single numerical value with negative values replaced by zero.
>   """
>   return np.maximum(0, x)
>
> def monte_carlo_y_k(d, k, num_samples=100000):
>     samples = []
>
>     for _ in tqdm(range(num_samples)):
>         x = np.random.randn(d)
>         y = x / np.linalg.norm(x)
>         y = relu(y)
>         y_sorted = np.sort(y)[::-1]
>         y_k = y_sorted[:k]
>         samples.append(1 / (y_k**2).sum() ** 0.5)
>
>     return np.mean(samples)
>
>
> class KERNRouter(nn.Module):
>     def __init__(
>         self,
>         initial_method="one",
>         total_expert=64,
>         top_k=8,
>         eps=1e-8,
>     ):
>         super().__init__()
>
>         if initial_method == "one":
>             self.scale_initial = 1
>         elif initial_method == "monte_carlo":
>             self.scale_initial = monte_carlo_y_k(total_expert, top_k)
>         self.scale = nn.Parameter(torch.ones(1))
>         self.eps = eps
>         self.activation = nn.ReLU()
>
>     def forward(self, x):
>         norm_x = x.norm(2, dim=-1, keepdim=True)
>         x_normed = x / (norm_x + self.eps)
>         x_normed=self.activation(x_normed)
>
>         return self.scale * self.scale_initial * x_normed
> ```
>
> Reference:
>
> [1] Ioffe, S., & Szegedy, C. (2015, June). Batch normalization: Accelerating deep network training by reducing internal covariate shift. In International conference on machine learning (pp. 448-456). pmlr.
>
> [2] Kim, D., Kim, W., & Kim, S. (2023). Tanh works better with asymmetry. Advances in Neural Information Processing Systems, 36, 12536-12554.

---

> ### Author Response · Authors · 2025-11-14
> **Response to Reviewer XdAq (Part 2/2)**
>
> **Q4: How does changing the order between normalization and activation (as seen in Eq. (4) vs. Eq. (3)) influence the theoretical interpretation or empirical behavior of the model? Has this difference been empirically tested or justified in the paper?**
>
> A4: **The following is the empirical performance with KERN-ReLU First and KERN**
>
>
> *The performance with loss metrics*
>  |MoE Router |  5K | 10K | 15K | 20K | 25K | 30K | 35K | 40K | 45K | 50K |
> |---|----|-----|-----|-----|-----|-----|-----|-----|-----|-----|
> | KERN-ReLU First  | 4.2228 | 3.8683 | 3.7954 | 3.6800 | 3.5978 | 3.5527 | 3.4795 | 3.3729 | 3.3647 | 3.3132
> | KERN  |  **4.2165** | **3.8650** | **3.7919** | **3.6724** | **3.5910** | **3.5503** | **3.4776** | **3.3707** | **3.3608** | **3.3080**
>
> **The KERN (Norm-ReLU) consistently improves the performance, compared to KERN-ReLU First (ReLU-Norm)**. If first relu and then norm, then 50% of the router logit will not participate in the computation flow, so that there is no gradient. We will further explain Equation 3 and 4 to avoid such ambiguity.
>
>
> **Q5: Can the proposed KERN router be extended beyond MoE architectures to serve as a general routing or attention mechanism in neural networks, similar to how Softmax is used broadly across architectures?**
>
> A5: **The KERN could be extended beyond MoE architectures to serve as a general routing or attention mechanism, similar to how Softmax is used broadly across architectures?**
>
> *The performance with loss metrics*
>
> |MoE Router | Attention Mechnism | 5K | 10K | 15K | 20K | 25K | 30K | 35K | 40K | 45K | 50K |
> |---|---|----|-----|-----|-----|-----|-----|-----|-----|-----|-----|
> | Softmax  | Softmax  | 4.3011 | 3.9318 | 3.8586 | 3.7394 | 3.6594 | 3.6185 | 3.5507 | 3.4459 | 3.4387 | 3.3882
> | KERN  | KERN |      4.3307 | 3.9108 | 3.8262 | 3.7052 | 3.6229 | 3.5790 | 3.5070 | 3.3998 | 3.3918 | 3.3422
> | KERN  | Softmax  | **4.2165** | **3.8650** | **3.7919** | **3.6724** | **3.5910** | **3.5503** | **3.4776** | **3.3707** | **3.3608** | **3.3080**
>
>
>
>
> **The proposed KERN could be extended to general routing or attention mechanisms**. Replace the Softmax with KERN, the neural network is still trainable.
>
> **Pure KERN (MoE with KERN and Attention with KERN) achieves better performance than Pure Softmax (MoE with softmax and Attention with softmax)**.
>
> **The MoE with KERN and Attention Softmax achieves the best performance.**
> **Softmax may be better when the token interacts with a different number of other embeddings**. In Attention with decoder-only mask, the token at position $i$ may interact with a total $i$ tokens, which is not fixed. **KERN may be better when the token interacts with a fixed number of other embeddings.** In MoE router, the input $x$ will  interact with $W \in R^{D \times K}$, which has $K$ embeddings.
>
>
> **If the response addresses your concerns, could you consider increasing the score?**

---

> > ### Author Response · Authors · 2025-11-19
> >
> > Dear Reviewer XdAq,
> >
> > Thank you for the support. We explain that the order of normalization and activation function (**updated revision for Equation 4 to eliminate the ambiguity**), add the comparison between KERN-ReLUFirst to the paper (**Appendix O**), and the additional experiment for KERN beyond MoE (e.g., replace softmax in attention).  **What else could we do to help consider improving the score? If there are no further concerns, could you consider improving the score?**

---

> ### Author Response · Authors · 2025-11-24
> **One Week Before Discussion End**
>
> Dear Reviewer XdAq,
>
> The response addresses the mentioned concerns, and there is about one week left for the discussion. **What else could we do to help consider improving the score? If there are no further concerns, could you consider improving the score? Thank you for the attention and engagement, and wish you a good day.**

---

> > ### Author Response · Authors · 2025-11-27
> > **The Discussion will End in 3 Working Days**
> >
> > Dear Reviewer XdAq,
> >
> > Hope this finds you well. **As the rebuttal will end in three working days, if the response addresses the concerns, could you consider to increase the score? Thank you for the attention**

---

### Official Review · Reviewer_qEoZ · 2025-10-30

**Soundness:** 1
**Presentation:** 2
**Contribution:** 1
**Rating:** 2
**Confidence:** 5

**Summary:**

This paper reinterprets MoE routing through the lens of the Nadaraya–Watson (NW) estimator, arguing that both FFNs and MoE can be seen as parametric NW regressors. Building on this view, the authors propose KERN (Kernel-Inspired Router with Normalization): a “FFN-style” router that replaces Softmax with a linear projection followed by L2-normalization, ReLU, and a learnable global scale, combined with standard Top-k selection. The motivation is to avoid exponential activations’ saturation, keep output scale stable, and improve expert utilization. Experiments from 520M→6.9B (active 125M→1.3B) on Books3/ArXiv and a 50B-token FineWeb-Edu pretrain report consistent loss gains and better zero-shot downstream accuracy versus Softmax/Sigmoid/Tanh routers and dense baselines.

**Strengths:**

- **Clear, unified view.** The NW formulation offers a simple mathematical template that maps FFN and MoE into the same “kernel+normalization” structure, making the proposed design choices easy to state and implement.
- **Low engineering overhead.** KERN is a drop-in change (ReLU+L2norm+γ) and the paper claims no substantive time cost relative to Softmax routing.
- **Consistent wins in the reported setup.** Across multiple model sizes and datasets, KERN slightly but repeatedly outperforms Softmax/Sigmoid/Tanh, including at longer contexts.

**Weaknesses:**

- **Limited novelty.** The proposed router is essentially a straightforward substitution of the probabilistic Softmax with a normalized ReLU gate, i.e., $\gamma \cdot ReLU(norm(xW)))$, in replace of $Softmax(xW)$. While the Nadaraya–Watson (NW) perspective is conceptually interesting, the paper mainly **instantiates** an NW-like form without analyzing whether NW’s assumptions (kernel choice, normalization, bandwidth/scale selection) align with MoE routing objectives (load balance, capacity control, calibration). The stated motivation—“Softmax/Sigmoid suffer gradient saturation/vanishing”—is asserted rather than demonstrated (e.g., no gradient-magnitude, activation-scale, or saturation diagnostics).
- **Insufficient comparison to closely related work.** Most components of the final design already exist in the literature, yet the paper primarily compares against vanilla Softmax/Sigmoid/Tanh routers, omitting stronger, conceptually adjacent baselines and ablations.
	- **ReLU routing:** \[1\] replaces Softmax with ReLU to obtain fully differentiable, dynamic routing.
    - **Logit normalization:** \[2\] normalizes routing logits to decouple expert-centroid magnitude from assignment.
    - **Output scaling:** Industrial implements like \[3,4\] adjusts the routing output scale to preserve training dynamics comparable to dense models.

- **Questionable empirical setup and reporting.** The reported gaps show Sigmoid performing markedly worse than Softmax—sometimes **larger** than the gap between Softmax-MoE and its dense counterpart—which is atypical relative to prior results \[2,5\] and suggests possible tuning or configuration mismatches. Critical details for effective MoE training are missing: the load-balancing objective (form and coefficient), capacity factor/dropless vs padded routing, post-Top-(k) renormalization (if any), and per-expert utilization/entropy or drop-rate statistics. Without these, it is difficult to assess stability, fairness of comparisons, or reproducibility of the claimed gains.

\[1\] ReMoE: Fully Differentiable Mixture-of-Experts with ReLU Routing https://arxiv.org/pdf/2412.14711
\[2\] On the Representation Collapse of Sparse Mixture of Experts https://arxiv.org/pdf/2204.09179
\[3\] DeepSeek-V3 Technical Report https://arxiv.org/abs/2412.19437
\[4\] Kimi K2: Open Agentic Intelligence https://arxiv.org/abs/2507.20534
\[5\] Auxiliary-Loss-Free Load Balancing Strategy for Mixture-of-Experts https://arxiv.org/pdf/2408.15664

**Questions:**

- How does the scale parameter $\gamma$ evolve during training (per layer and across settings)?
- Do gradient vanishing and saturation actually occur with Softmax and Sigmoid routers, and does KERN empirically mitigate them?
- Does KERN improve expert load balancing?
- How does KERN behave in MoE architectures with shared experts?

---

> ### Author Response · Authors · 2025-11-14
> **Response to Reviewer qEoZ (Part 1/4)**
>
> Dear Reviewer   qEoZ,
>
> Thank you for the comments. We will address the concerns below.
>
> **Q1: The novelty of this work**
>
> A1: **The main contribution is: We interpret the FFN and MoE with Nadaraya-Watson kernel regression, and we empirically validate that the FFN and MoE could both benefit from the ReLU activation function**
>
> * **We are the first ones to connect with FFN and MoE with Nadaraya-Watson Kernel Regression**
> * **Before this work, the community wrongly believed that the more activated parameters, the better the performance. In Appendix J, we empirically validate that the MoE could benefit from the activation function such as ReLU**
>
>
> **Q2: While the Nadaraya–Watson (NW) perspective is conceptually interesting, the paper mainly instantiates an NW-like form without analyzing whether NW’s assumptions (kernel choice, normalization, bandwidth/scale selection) align with MoE routing objectives**
>
> A2: **We regarded the MoE with the Nadaraya-Watson Kernel Regression formulation, which is not related to the kernel choice and so on**
>
> **Nadaraya-Watson Kernel Regression**: $f_{NW}(x) = \sum_{i=1}^N \frac{K(x, x_i)}{\sum_{j=1}^N K(x, x_j)} y_i$
>
> **Mixture-of-Experts**:  $\text{MoE}(x) = \sum_{m=1}^M \hat{g}_m(x) E_m(x),$
>
> **Mixture-of-Experts Routing**: $g_m(x) = \frac{K(\langle w_m, x)}{\sum_{j=1}^M K(\langle w_j, x \rangle)}.$
>
> **MoE is the Nadaraya-Watson Kernel Regression, while the Mixture-of-Experts Routing is not. The Nadaraya-Kernel Regression could choice any suitable kernel, such as softmax kernel, sigmoid kernel, uniform kernel and so on. The MoE router could also choose any potential kernel, such as softmax, sigmoid, tanh and so on. And the MoE could also remove the normalization part $\sum_{j=1}^M K(\langle w_j, x \rangle)$.**
>
> **Q3: Compare to more baselines**
>
> A3:  We compare the KERN with ReLU routing and logit normalization in the following. Compared with the baseline, the KERN achieves the best performance.
>
> *The performance with loss metrics*
> |MoE Router |  5K | 10K | 15K | 20K | 25K | 30K | 35K | 40K | 45K | 50K |
> |---|----|-----|-----|-----|-----|-----|-----|-----|-----|-----|
> | ReLU  |4.2496 | 3.8987 | 3.8241 | 3.7018 | 3.6203 | 3.5791 | 3.5099 | 3.4027 | 3.3955 | 3.3441
> | logit normalization  | 4.2743 | 3.8858 | 3.8044 | 3.6814 | 3.5999 | 3.5582 | 3.4857 | 3.3801 | 3.3718 | 3.3194
> | output scale  | 4.2288 | 3.8756 | 3.7994 | 3.6801 | 3.6007 | 3.5594 | 3.4853 | 3.3791 | 3.3729 | 3.3220
> | KERN  |  **4.2165** | **3.8650** | **3.7919** | **3.6724** | **3.5910** | **3.5503** | **3.4776** | **3.3707** | **3.3608** | **3.3080**

---

> ### Author Response · Authors · 2025-11-14
> **Response to Reviewer qEoZ (Part 2/4)**
>
> **Q4: reported gaps show Sigmoid performing markedly worse than Softmax—sometimes larger than the gap between Softmax-MoE and its dense counterpart—which is atypical relative to prior results**
>
> A4: **This is the misunderstanding for the experiment, while our conclusion is aligned with the previous works: sigmoid  is better than softmax**
>
> * **It is clearly presented that Sigmoid is better than Softmax in Section 4.1 Comparison with Baseline.** In section 4.1, the Sigmoid (green line) always achieves better performance than the Softmax (orange line) so that Sigmoid loss is lower than Softmax. The observation is the same with different datasets Arxiv or Books3, and different length 512, 1024 and 2048.
>
> * **It is clearly presented that Sigmoid is better than Softmax in Section 4.2 Large Model Performance.** Sigmoid consistently achieves the better performance than Softmax at every scale. At 350M parameters, Sigmoid loss of 3.2279 is lower than that of the comparable Softmax (3.2709) and Dense (3.3500) models. This lead is extended at the 1.3B scale, where Sigmoid loss of  3.1388 significantly outperforms the Softmax (3.1814) and Dense (3.2219) results.
>
> * **It is clearly presented that Sigmoid is better than Softmax in Section 4.3 The Effect of Granularity.** with 256 experts, 8 active experts, and an expert intermediate size is 96, the Softmax achieves 4.3139 loss at evaluation step 5K and 3.4150 loss at evaluation step 50K. The Sigmoid and Tanh achieve 4.2924 loss and 4.2435 loss at the evaluation step 5K, and Sigmoid and Tanh achieve 3.3302 loss and 3.3276 loss at the evaluation step 50K. The KERN achieves the best performance 4.2294 loss at evaluation step 5K and 3.2962 loss at evaluation step 50K.
>
> * **It is clearly presented that Sigmoid is better than Softmax in Section 4.4 The Effect of Sparsity.** The Softmax achieves 4.3059 loss at evaluation step 5K and 3.3761 loss at evaluation step 50K. The Sigmoid and Tanh achieve 4.2924 loss and 4.2435 loss at the evaluation step 5K, and Sigmoid and Tanh achieve 3.2760 loss and 3.2972 loss at the evaluation step 50K. The KERN achieves the best performance 4.1926 loss at evaluation step 5K and 3.2672 loss at evaluation step 50K.
>
> * **It is clearly presented that Sigmoid is better than Softmax in Section 4.5 The Performance on Large-Scale Pretrain Dataset.** At a 520M model size (125M active), KERN achieves an average performance of 52.14, surpassing Dense (48.51), Softmax (49.88), Tanh (51.53), and Sigmoid (51.80). With a 1.7B model size (350M active), it scores 55.13, outperforming Dense (51.05), Softmax (52.46), Tanh (54.18), and Sigmoid (54.72). Similarly, at 6.9B (1.3B active), it reaches 58.88, exceeding Dense (56.11), Softmax (56.49), Tanh (58.04), and Sigmoid (58.55).

---

> ### Author Response · Authors · 2025-11-14
> **Response to Reviewer qEoZ (Part 3/4)**
>
> **Q5: The details of the training**
>
> A5: **We have clearly presented the training details in Appendix D Model Configuration, and our training setting follows the MoE++ (ICLR 2025 Oral Paper) [1]**, with top-k routing, dropless MoE. The softmax and router follow the setting of DeepSeek-V3 and Kimi K2, while the softmax is pre-top-k normalized and the sigmoid is post-top-k normalized.
>
> | | | 125M | | 350M |
> |---|---|---|---|---|
> | Training sequence length | | $512$ | | $512$ |
> | Batch size | | 32 $\times$ 8 | | 32 $\times$ 8 |
> | Number of iterations | | $50$k | | $50$k |
> | Dropout prob. | | $0.0$ | | $0.0$ |
> | Attention dropout prob. | | $0.0$ | | $0.0$ |
> | Attention head | | 12 | | 16 |
> | Feature dimension | | 768 | | 1024 |
> | Layer number | | 12 | | 24 |
> | Optimizer | | Adam | | Adam |
> | Optimizer parameter betas | | [0.9, 0.95] | | [0.9, 0.95] |
> | Learning rate | | $6\mathrm{e}-4$ | | $3\mathrm{e}-4$ |
> | Precision | | float16 | | float16 |
> | Total Expert Number | | 64 | | 64 |
> | Active Expert Number | | 8 | | 8 |
>
> **Q6: How does the scale parameter  evolve during training (per layer and across settings)?**
>
>  A6: For different layler, usually the scale parameter will be gradually increased and achieve the largest value at the middle layer, and then decrease. Usually, the first and second layer scales may be larger or smaller to the initialization value, while the middle layer is larger than the initialization value.
>
>  *The scale value*
> |layer | 1 | 2 | 3 | 4| 5 | 6 | 7| 8 | 9 | 10 | 11 | 12 |
> |---|---|----|-----|-----|-----|-----|-----|-----|-----|-----|-----| -----|
>  Scale|1.0703 |1.6764| 0.8281| 1.9688| 2.4844| 2.5469| 2.6719| 3.1875| 3.4291| 3.6250| 2.8750|  1.3906|
>
>  **Q7: Do gradient vanishing and saturation actually occur with Softmax and Sigmoid routers, and does KERN empirically mitigate them?**
>
>  A7: Yes, we observe that the softmax and sigmoid have gradient vanishing. For example, the maximum of the softmax value and sigmoid value could be close to 1, so that the gradient is almost vanishing.
>
> ```
> Example: input [-2 -3 -1 2 3 10] :
> KERN (L2Norm): [-0.177  -0.266 -0.088  0.177   0.266  0.887]
> Sigmoid: [0.119 0.0474 0.268 0.880 0.952 0.999]
> ###The router logit will not have a gradient for $w_6$
> Softmax: [6.136e-06 2.257e-06 1.668e-05 3.350e-04 9.107e-04 9.987e-01]
> ######The graient is almost vanish for $w_1$ to $w_6$
>
> ReLU & choose top-2 for KERN: [0 0 0 0 0.266  0.887]
> #### $w_1$ to $w_6$ all have gradient because of L2 normalization.
> #### Note that whether there is ReLU or not, the choice of top-2 will not change the final result because usually the ReLU only masks 50%, while top-2 masks 4/6=66.7%.
> #### When the top-k for K/M>50%, adding ReLU may improve the performance, which is similar to FFN (Appendix J).
>
> choose top-2 for Sigmoid: [0 0 0 0 0.952 0.999]
> #### the sigmoid will NOT have gradient for $w_1$ to $w_4$ because only 2 values are selected. And $w_6$ also does not have a gradient that vanishes because 0.999 is too large.
>
> choose top-2 for Softmax: [0 0 0 0 9.107e-04 9.987e-01]
> ### the softmax will NOT have a gradient for $w_1$ to $w_6$ because one value is very large.
> ```
>
>
>  **Q8: Does KERN improve expert load balancing?**
>
>  A8: **KERN improves expert load balance.**
>
>  We use the following to calculate the expert load: calculate the expert frequency and compare with the uniform distribution.
>
>  *The KL loss between expert selection frequency and uniform distribution
>
>  *The KL loss for expert load frequency and uniform distribution*
>  |Method | Softmax|Sigmoid | KERN |
>  |---|---|----|----|
>  |KL Loss Mean| 0.9483|0.3162|**0.2947**
>   |KL Loss Std| 0.1997|0.2207|**0.1896**
>
>  The softmax achieves the highest KL loss, suggesting that its expert load is not balanced. The Sigmoid achieves and KERN lower KL loss than Softmax, suggesting that Sigmoid and KERN have better expert load.

---

> ### Author Response · Authors · 2025-11-14
> **Response to Reviewer qEoZ (Part 4/4)**
>
> **Q9: How does KERN behave in MoE architectures with shared experts?**
>
>  A9: The following is the KERN with 2 shared experts, following the DeepSeek setting [2]. **The KERN achieves the best performance with shared experts.**
>
>
>  *The performance with loss metrics*
>  |MoE Router |  5K | 10K | 15K | 20K | 25K | 30K | 35K | 40K | 45K | 50K |
> |---|----|-----|-----|-----|-----|-----|-----|-----|-----|-----|
> | Softmax  |4.2769 | 3.9061 | 3.8385 | 3.7232 | 3.6476 | 3.6057 | 3.5372 | 3.4318 | 3.4267 | 3.3744
> | Sigmoid  | 4.2555 | 3.8817 | 3.8052 | 3.6864 | 3.6088 | 3.5641 | 3.4920 | 3.3841 | 3.3770 | 3.3251
> | Tanh  | 4.2705 | 3.8972 | 3.8185 | 3.6941 | 3.6139 | 3.5706 | 3.5002 | 3.3910 | 3.3810 | 3.3301
> | KERN  |  **4.2310** |  **3.8586** |  **3.7831** |  **3.6634**|  **3.5873** |  **3.5448** |  **3.4737** |  **3.3660** |  **3.3571** |  **3.3043**
>
>
> Reference:
>
> [1] Jin, P., Zhu, B., Yuan, L., & Yan, S. (2024). Moe++: Accelerating mixture-of-experts methods with zero-computation experts. arXiv preprint arXiv:2410.07348.
>
> [2] Liu, A., Feng, B., Wang, B., Wang, B., Liu, B., Zhao, C., ... & Xu, Z. (2024). Deepseek-v2: A strong, economical, and efficient mixture-of-experts language model. arXiv preprint arXiv:2405.04434.
>
>
> **If the response addresses your concerns, could you consider increasing the score?**

---

> > ### Comment · Reviewer_qEoZ · 2025-11-17
> > **Reply to the Authors**
> >
> > The rebuttal does not adequately address my main concerns. I will maintain my score of 2.
> >
> > 1. The authors claim to be the first to connect MoE with NW regression. However, this connection does not yield any substantial new insight. The title promises “understanding MoE with NW kernel,” but the paper essentially only notes that MoE and NW share the same formulation, without seriously exploring what this perspective brings either theoretically or empirically.
> >
> > 2. I explicitly suggested that the authors compare their method with recent works on improving MoE, since only vanilla routing baselines are included. However, the revision only reports ablations of the proposed method itself, without adding any direct and fair comparisons to these stronger baselines.
> >
> > 3. There are prior works reporting that Sigmoid routing can outperform Softmax, but not to the extent shown in the experiments here. In this paper, the loss gap between Sigmoid MoE and Softmax MoE is even larger than that between Softmax MoE and dense models, which seems highly implausible and calls for more careful analysis and explanation.
> >
> > 4. The detailed configuration of load balancing is still not provided (eg. $\alpha$ in Switch load balancing loss), which it is crucial for stabilizing MoE training and for interpreting the reported gains.

---

> ### Author Response · Authors · 2025-11-17
> **Response to Reviewer qEoZ (Part 1/2)**
>
> Dear Reviewer qEoZ,
>
> Thank you for the comment. We will address your concerns below.
>
> **Q1: The title promises “understanding MoE with NW kernel,” but the paper essentially only notes that MoE and NW share the same formulation, without seriously exploring what this perspective brings either theoretically**
>
> A1: **Insight:**
> *  **Theoretical insight: 1) provides a unified mathematical perspective linking MoE, FFN, and classical kernel regression; 2) Motivated by this insight, brings the success of FFN to MoE**
> * **Empirical Insight: We challenge the common belief: the more activated parameters of MoE, the better the performance. In Appendix J, the MoE is the same as FFN. Adding ReLU (reducing the activated parameters) could improve the performance if there are too many activated parameters**
>
>
> The following is the theoretical analysis for MoE, FFN and  classical kernel regression.
>
> After training, the inference procedure of the MoE model is mathematically equivalent to performing NW regression. We elaborate on this connection below.
> Consider an MoE model with $K$ experts (e.g., language models). For a given input context $X$, each expert produces an output $Y_1, Y_2, \cdots, Y_K$. The router assigns a weight to each expert through a routing function $g_m(X)$. This routing function can be interpreted as a kernel function $K(w_m, X)$, where $w_m$ denotes the expert-specific router parameters.
>
> For a softmax router, we have $K(w_m, X) = \exp(<w_m, X>)$, which is a symmetric kernel function in $w_m$ and $X$. Thus, the router measures the similarity between the input token embedding $X$ and the router vector $w_m$. The final MoE output is simply a similarity-weighted aggregation of the expert outputs $Y_1, Y_2, \cdots, Y_K$. Hence, at inference, a trained MoE implements exactly the form of NW regression, where experts correspond to the "samples," router vectors serve as kernel centers, and the router functions act as kernel evaluations.
>
>
> In Section 3.2, we perform a similar analysis for FFNs, showing that FFNs are also mathematically equivalent to NW regression. In Equation (3), each parameter vector in the final (output) layer represents an output value to be aggregated. The weight vectors $w_m$ in the penultimate layer determine similarity scores between $w_m$ and $X$ (or more generally, $\Phi(x)$ due to multiple preceding layers). For an $L$-layer FFN, the first $L-2$ layers serve as a preprocessing map $\Phi$. The $(L-1)$-th layer computes similarity scores (i.e., kernel evaluations) between $w_m$ and $\Phi(x)$. The final layer then aggregates the outputs according to these similarity scores, again matching the structure of NW regression.
>
> Together, these analyses demonstrate that both MoE and FFN architectures are special cases of the NW regression framework.
>
> Motivated by this insight, we further propose a more general and better-suited MoE configuration. Since MoE is predominantly used in deep learning contexts, where classical kernel choices may not be ideal, we draw inspiration from the success of FFN architectures. We incorporate FFN-style kernel functions into the MoE router, which we argue leads to kernel functions that are more aligned with and well-suited to modern deep learning tasks.
>
>
> **Q2: However, the revision only reports ablations of the proposed method itself, without adding any direct and fair comparisons to these stronger baselines.**
>
> A2: **This may be misunderstanding. The revision is not the ablation of the proposed method, but the direct and fair comparsion to these stronger baselines**
> * ReLU routing: [1] replaces Softmax with ReLU to obtain fully differentiable, dynamic routing. **Therefore, we follow the work to use the ReLU routing.**
> * Logit normalization: [2] normalizes routing logits to decouple expert-centroid magnitude from assignment. **Therefore, we follow the work use the Logit normalization and sigmoid, which achieves the best performance in [2].**
> * Output scaling: Industrial implements like [3,4] adjusts the routing output scale to preserve training dynamics comparable to dense model. **Therefore, we follow the work to use the output scaling and sigmoid, which is used in DeepSeekV3 and Kimi K2.**
>
> *The performance with loss metrics*
> |MoE Router | Implementation| 5K | 10K | 15K | 20K | 25K | 30K | 35K | 40K | 45K | 50K |
> |---|----|-----|-----|-----|-----|-----|-----|-----|-----|-----|-----|
> | ReLU  |ReLU(xW)|4.2496 | 3.8987 | 3.8241 | 3.7018 | 3.6203 | 3.5791 | 3.5099 | 3.4027 | 3.3955 | 3.3441
> | logit normalization  |Sigmoid($xw_i$ / (Norm(x)*Norm($w_i$))| 4.2743 | 3.8858 | 3.8044 | 3.6814 | 3.5999 | 3.5582 | 3.4857 | 3.3801 | 3.3718 | 3.3194
> | output scale  | Sigmoid(xW)*scale|4.2288 | 3.8756 | 3.7994 | 3.6801 | 3.6007 | 3.5594 | 3.4853 | 3.3791 | 3.3729 | 3.3220
> | KERN  | Equation 7|  **4.2165** | **3.8650** | **3.7919** | **3.6724** | **3.5910** | **3.5503** | **3.4776** | **3.3707** | **3.3608** | **3.3080**

---

> ### Author Response · Authors · 2025-11-17
> **Response to Reviewer qEoZ (Part 2/2)**
>
> **Q3: In this paper, the loss gap between Sigmoid MoE and Softmax MoE is even larger than that between Softmax MoE and dense models, which seems highly implausible and calls for more careful analysis and explanation.**
>
> A3: **The sigmoid is consistently better than softmax. However, it is only potential that the Sotmax-Dense gap  is smaller than the Sigmoid-Softmax but not guaranteed.**
>
>
> *The validation loss on Books3 dataset*
> | Model                |  5K   |  10K  |  15K  |  20K  |  25K  |  30K  |  35K  |  40K  |  45K  |  50K  |
> | :------------------- | :---- | :---- | :---- | :---- | :---- | :---- | :---- | :---- | :---- | :---- |
> |Dense|3.87 | 3.67 | 3.56   |  3.49   |  3.43   |  3.38   |  3.32   |  3.27   |  3.26   |  3.24|
> | **8 of 64 Active Expert** |
> | Softmax              | 4.02  | 3.68  | 3.55  | 3.47  | 3.40  | 3.34  | 3.27  | 3.22  | 3.20  | 3.17 (reduce 0.07, compared to Dense)  |
> |Sigmoid|3.96  | 3.63  | 3.49  | 3.40  | 3.33  | 3.28  | 3.20  | 3.15  | 3.13  | 3.10 (reduce 0.07, compared to Softmax)  |
> | **64 of 64 Active Expert** |
> | Softmax              | 3.99  | 3.66  | 3.51  | 3.43  | 3.35  | 3.30  | 3.22  | 3.16  | 3.15  | 3.12 (reduce 0.12, compared to Dense)  |
> | Sigmoid              | 3.99  | 3.64  | 3.50  | 3.41  | 3.34  | 3.28  | 3.20  | 3.15  | 3.13  | 3.10 (reduce 0.02, compared to Sigmoid)  |
>
> *Performance from ReMoE*
> | Model                | Router| ARC-c | ARC-e | BoolQ  |HellaSwag | LAMBADA | PIQA | RACE
> | :------------------- | :---- | :---- | :---- | :---- | :---- | :---- | :---- | :---- |
> |Dense|19.45 | 43.35 | 54.40 | 28.61 | 31.09 | 61.97 | 28.52 | 38.20
> |SparseMixer-v2| Softmax              | 19.80 | 46.72 | 45.96 | 30.24 | 34.12  |62.89 | 29.00 | 38.39 (improve 0.19, compared to Dense)  |
> | ReMoE| ReLU             | 20.22 | 46.68  | 54.16  | 30.26  | 35.94  | 63.55  | 29.38  | 40.03 (improve 1.83, compared to Sigmoid)  |
>
> **As shown in ReMoE, the performance gap between one variant (e.g. SparseMixer-v2) and Dense could be significantly smaller than the performance gap between the variant (e.g. SparseMixer-v2) and another variant (e.g. ReMoE). Therefore, it is possible that Sotmax-Dense gap is smaller than Sigmoid-Softmax.**
>
>
>
> **Q4: The detailed configuration of load balancing is still not provided (eg.
>  in Switch load balancing loss), which it is crucial for stabilizing MoE training and for interpreting the reported gains.**
>
>  A4: **Following the previous work suggestion (e.g., DeepSeek V3), we further validate our method with the method auxiliary-loss-free load (default) and sinkhorn load method (Appendix M) [1,5,6,7,8,9,10]. We will release our code for production. And we have high confidence to say that all our experiments are fair and reproducible. We have presented the implementation in Appendix J.**
>
> **If possible, the reviewer could use our proposed method KERN to replace the sigmoid or softmax router to conduct experiments for validation, which takes less than 1 minute to finish the replacement. We have confidence to say that our proposed method will achieve the best performance.**
>
>
> Reference:
>
> [1]  Wang, Z., Zhu, J., & Chen, J. (2024). Remoe: Fully differentiable mixture-of-experts with relu routing. arXiv preprint arXiv:2412.14711.
>
> [2] Chi, Z., Dong, L., Huang, S., Dai, D., Ma, S., Patra, B., ... & Wei, F. (2022). On the representation collapse of sparse mixture of experts. Advances in Neural Information Processing Systems, 35, 34600-34613.
>
> [3] Liu, A., Feng, B., Xue, B., Wang, B., Wu, B., Lu, C., ... & Piao, Y. (2024). Deepseek-v3 technical report. arXiv preprint arXiv:2412.19437.
>
> [4]  Team, K., Bai, Y., Bao, Y., Chen, G., Chen, J., Chen, N., ... & Zhang, H. (2025). Kimi k2: Open agentic intelligence. arXiv preprint arXiv:2507.20534.
>
> [5] Zeng, Z., Guo, Q., Fei, Z., Yin, Z., Zhou, Y., Li, L., ... & Qiu, X. (2024, November). Turn Waste into Worth: Rectifying Top-k Router of MoE. In Proceedings of the 2024 Conference on Empirical Methods in Natural Language Processing (pp. 13305-13320).
>
> [6] Liu, T., Blondel, M., Ruiz, C. R., & Puigcerver, J. Routers in Vision Mixture of Experts: An Empirical Study. Transactions on Machine Learning Research.
>
> [7] Clark, A., de Las Casas, D., Guy, A., Mensch, A., Paganini, M., Hoffmann, J., ... & Simonyan, K. (2022, June). Unified scaling laws for routed language models. In International conference on machine learning (pp. 4057-4086). PMLR.
>
> [8] Wang, X., Wang, S., Ding, Y., Li, Y., Wu, W., Rong, Y., ... & Tang, J. (2024). State space model for new-generation network alternative to transformers: A survey. arXiv preprint arXiv:2404.09516.
>
> [9] Wang, L., Gao, H., Zhao, C., Sun, X., & Dai, D. (2024). Auxiliary-loss-free load balancing strategy for mixture-of-experts. arXiv preprint arXiv:2408.15664.
>
> [10] Liu, A., Feng, B., Xue, B., Wang, B., Wu, B., Lu, C., ... & Piao, Y. (2024). Deepseek-v3 technical report. arXiv preprint arXiv:2412.19437.

---

> > ### Author Response · Authors · 2025-11-20
> >
> > Dear Reviewer qEoZ,
> >
> > Thank you for the comment. We explain the theoretical and empirical insights, the comparison with stronger baselines, the performance of sigmoid and softmax, and the experiment setting details.. **What else could we do to help consider improving the score? If there are no further concerns, could you consider improving the score?**

---

> > > ### Author Response · Authors · 2025-11-24
> > > **One Week Before Discussion End**
> > >
> > > Dear Reviewer qEoZ,
> > >
> > > The response addresses the major concerns, and there is about one week left for the discussion. **What else could we do to help consider improving the score? If there are no further concerns, could you consider improving the score? Thank you for the attention and engagement, and wish you a good day.**

---

> > ### Comment · Reviewer_qEoZ · 2025-11-25
> >
> > The reviewer has carefully checked the revised manuscript and the rebuttal, and has decided to maintain the original rating of 2.
> >
> > **What this work essentially does:**
> > -  Demonstrates that NW and FFN/MoE share a similar mathematical form.
> > -  Modifies the MoE router from $Softmax(XW)$ to $\gamma \cdot ReLU(L2norm(XW))$.
> > -  Claims improved empirical performance.
> >
> > **In response to the authors:**
> > - Novelty claimed by the authors:
> > 	-  *"The first one to connect FFN/MoE with NW", "Provides a unified mathematical perspective linking MoE, FFN, and classical kernel regression"*: The connection itself is not inherently valuable. The key question is **what insight this connection offers**. I do not challenge *"MoE and FFN architectures are special cases of the NW regression framework"*. However, how does this observation inform the design of better MoE? If the reasoning is merely *"classical kernel choices may not be ideal in NW regression"* -> "classical activation function may not be ideal in MoE", then the NW perspective becomes uninformative. It is already well understood that changing the activation function can yield better results under certain experimental settings, without invoking the NW framework. If the authors intend to position the paper as proposing a **better activation function for MoE routing**, as suggested by the experiments, the motivation should be made more rigorous: explain the intuition behind this activation choice and include stronger stress tests. For an academic contribution, comparisons with well-established activation functions such as GELU \[1\], SiLU \[2\], and GLU \[3\] are also necessary.
> > 	-  *"Challenging the common belief that more activated parameters brings better performance"*: This is a strong claim. If the authors consider it a notable contribution, the setup and evidence should be clearly described in the main text rather than hidden in the appendix.
> > 	- *"Brings the success of FFN to MoE"*: This statement is vague. The method simply replaces the router activation function. It is unclear what aspect of “FFN success” is being transferred to MoE, which is already a dynamic sparse variant of FFN.
> > - Experiment setup:
> > 	- I kindly remind the authors, for the third time, to provide detailed configurations for **load balancing**. If an auxiliary balancing loss is used, specify its coefficient. If the DeepSeek expert bias is applied, describe its update step. Additionally, the reported KL loss for Softmax (0.94) seems abnormally large, raising doubts about the correctness of the balancing configuration.
> > 	-  *"The revision is the direct and fair comparison to the stronger baselines"*: ReMoE\[4\] does not simply replace Softmax with ReLU, it also removes TopK and introduces sparsity regularization to make routing differentiable. X-MoE\[5\] does not simply apply L2 norm on $XW$, it also introduces dimension reduction and frozen routing to encourage uniform expert utilization and prevent routing collapse.
> > -  Experiment results:
> > 	- *"our conclusion is aligned with the previous works: sigmoid is better than softmax"*: This is partially true. Prior studies have indeed observed such trends, but the gap reported here is unusually large. Current industrial-scale MoE models typically employ softmax routing, and can beat a dense models of 3-4x times more active parameters. In contrast, the paper's results show that softmax-routed MoE (6.9B total, 1.3B active) is only **slighted** better than dense model (1.3B) with the same active parameters.
> >
> > |                    | Metric | Data Source | Dense | Softmax | Sigmoid | Proposed Method |
> > | ------------------ | ------ | ----------- | ----- | ------- | ------- | --------------- |
> > | X-MoE\[5\]         | Acc    | Table 1     | 61.4  | 63.8    | 64.1    | 65.3            |
> > | ReMoE\[4\]         | Acc    | Table 2     | 38.2  | 39.67   | \\      | 40.03           |
> > | Aux-Loss-Free\[6\] | PPL    | Figure 7    | \\    | ~9.60   | ~9.52   | ~9.50           |
> > | This work          | Acc    | Table 1     | 56.11 | 56.49   | 58.55   | 58.88           |
> >
> >
> > Minor problems:
> > - Page 5, Line 218, *"recent work has shown that replacing the Softmax with a Sigmoid router yields improved performance"*, please add citations.
> > - Typos:
> > 	- Page 3, Line 148, FNN -> FFN
> > 	- Page 9, Line 453, WinoGrande -> WinoGrade
> > 	- Page 19, Line 1003, withour -> without
> >
> > [1]Gaussian Error Linear Units (GELUs)
> >
> > [2]Sigmoid-Weighted Linear Units for Neural Network Function Approximation in Reinforcement Learning
> >
> > [3]GLU Variants Improve Transformer
> >
> > [4]ReMoE: Fully Differentiable Mixture-of-Experts with ReLU Routing
> >
> > [5]On the Representation Collapse of Sparse Mixture of Experts
> >
> > [6]AUXILIARY-LOSS-FREE LOAD BALANCING STRATEGY FOR MIXTURE-OF-EXPERTS

---

> ### Author Response · Authors · 2025-11-26
> **Response to  Reviewer qEoZ (Part 1/2)**
>
> Dear Reviewer qEoZ,
>
> Thank you for the comment, we will address the concerns below.
>
> **Q1: The insight**
>
> A1: The key insight in connecting MoE, FFN, and NW regression lies in narrowing the search space of router functions. In general, kernel functions can vary widely, which often leads to inefficient experimentation and suboptimal router selection in MoE. Since the appropriate kernel function depends on both the task and the data, bridging MoE with FFN through their shared connection to NW regression provides a useful perspective. Specifically, this motivates restricting the router function search space to architecture and activation functions commonly adopted in FFNs, given their architectural similarity and relevance to deep learning tasks. Prior evaluations have consistently shown that certain     architecture and activation functions are particularly well-suited for deep learning. By extension, we conjecture that MoE, as a deep learning framework, can also benefit from leveraging these FFN  architecture and  activation functions.
>
>
> **Q2: The MoE with gelu, silu, glu**
>
> A2: The performance of MoE with gelu, silu, glu.
>
> |MoE Router | 5K | 10K | 15K | 20K | 25K | 30K | 35K | 40K | 45K | 50K |
> |---|----|-----|-----|-----|-----|-----|-----|-----|-----|-----
> | GELU  | 4.2475 | 3.8951 | 3.8227 | 3.6998 | 3.6207 | 3.5755 | 3.5045 | 3.4001 | 3.3922 | 3.3393 |
> | SiLU  | 4.2438 | 3.8961 | 3.8193 | 3.6979 | 3.6180 | 3.5744 | 3.5050 | 3.3965 | 3.3923 | 3.3385 |
> | GLU  | 4.2455 | 3.8916 | 3.8229 | 3.7043 | 3.6233 | 3.5833 | 3.5143 | 3.4095 | 3.4021 | 3.3508 |
> | KERN  |   **4.2165** | **3.8650** | **3.7919** | **3.6724** | **3.5910** | **3.5503** | **3.4776** | **3.3707** | **3.3608** | **3.3080**
>
> **Q3: "Challenging the common belief that more activated parameters brings better performance": This is a strong claim. If the authors consider it a notable contribution, the setup and evidence should be clearly described in the main text**
>
> A3: Thank you for the comment. We revise the manuscript to highlight the Appendix J.
>
> **Q4: "Brings the success of FFN to MoE": This statement is vague. The method simply replaces the router activation function. It is unclear what aspect of “FFN success” is being transferred to MoE, which is already a dynamic sparse variant of FFN.**
>
>
> A4: Thank you for pointing this out. Our intention is to highlight how the structural relationship among FFNs, MoEs, and NW estimators provides a practical way to narrow down the design space of router activation functions.
>
> In the NW formulation, the kernel function is highly flexible. In theory, many different kernels can be used, and the choice depends on the specific task and data distribution. However, once we observe that both FFNs and MoEs can be interpreted as special cases of NW estimators, this structural similarity offers strong guidance for selecting an appropriate kernel. In particular, FFNs implicitly correspond to a specific class of kernel (activation) functions that are known to work well in deep neural network settings. By using this connection, we can restrict our search to the “FFN-style” kernels that are already empirically validated in large-scale deep learning, rather than exploring the full and much broader NW kernel space.
>
> To our knowledge, ours is the first work that explicitly links MoE, FFN, and NW, and uses this relationship to motivate the choice of the router activation function. This perspective allows us to justify why an FFN-inspired kernel is a reasonable and effective choice for MoE, providing a motivation rather than an ad-hoc design decision.
>
> We will revise the manuscript to clarify this point and avoid ambiguous claims.
>
>
>
>
> **Q5: If the DeepSeek expert bias is applied, describe its update step**
>
> A5: The update rate $\mu$ is set to 1e-6 for the experiment. The KL loss of softmax could decrease with larger $\mu$. The following is the implementation.
>
> |MoE Router | $\mu$|5K | 10K | 15K | 20K | 25K | 30K | 35K | 40K | 45K | 50K |
> |---|----|-----|-----|-----|-----|-----|-----|-----|-----|-----|-----|
> | KERN  |1e-3  （default by DeepSeek)|4.2203 | 3.8672 | 3.7929 | 3.6724 | 3.5947 | 3.5519 | 3.4784 | 3.3707 | 3.3610 | 3.3093 |
> | KERN  | 1e-6|**4.2165** | **3.8650** | **3.7919** | **3.6724** | **3.5910** | **3.5503** | **3.4776** | **3.3707** | **3.3608** | **3.3080**
>
> |MoE Router | $\mu$|KL Loss |
> |---|----|-----|
> | Softmax  |1e-3|0.9277|
> | Softmax  |1e-6| 0.9836|
>
> ```python
>
> #############DeepSeek Aux-Loss-Free Begin
> expert_counts = torch.bincount(topk_idx.flatten(),
>                                minlength=model.moe_layer.num_experts)
>
> # adjust biases for experts
> avg_count = expert_counts.float().mean()
> for i, count in enumerate(expert_counts):
>     error = avg_count - count.float()
>     model.moe_layer.expert_biases.data[i] += update_rate * torch.sign(error)
> #############DeepSeek Aux-Loss-Free End

---

> ### Author Response · Authors · 2025-11-26
> **Response to Reviewer qEoZ (Part 2/2)**
>
> **Q6: The revision is the direct and fair comparison to the stronger baselines**
>
> A6: **The following is the comparison with ReMoE and X-MoE.**
>
>
> *The loss on the wikitext dataset*
> |Method | Router Function|Load Balance|$\lambda_0$|$\alpha$|Loss|
> |---|----|-----|-----|-----|-----|
> | ReLU  |ReLU|Top-K|N/A | N/A|2.8503|
> | KERN  | KERN|Top-K|N/A | N/A | 2.7629|
> | ReLU  |ReLU|Sparsity Regularization|1e-12 | 1.2| 2.8138|
> KERN| ReLU|Sparsity Regularization|1e-16 | 1.2|2.7476|
> | ReLU  |ReLU|Sparsity Regularization|1e-8 | 1.1| 2.8353|
> KERN| ReLU|Sparsity Regularization|1e-8 | 1.1|2.7420|
>
>
> *The loss of X-MoE and KERN. **The X-MoE freeze is applied after training step 45K, as the X-MoE freeze is used only for the SFT stage, while this work focuses on the pretrain stage***
> |MoE Router | 5K | 10K | 15K | 20K | 25K | 30K | 35K | 40K | 45K | 50K |
> |---|----|-----|-----|-----|-----|-----|-----|-----|-----|-----|
> | X-MoE (normalization+dimension reduction)  |4.2970 | 3.9028 | 3.8252 | 3.7093 | 3.6302 | 3.5898 | 3.5185 | 3.4134 | 3.4063 | 3.3553 |
> | X-MoE (normalization+dimension reduction+freeze)  |4.2970 | 3.9028 | 3.8252 | 3.7093 | 3.6302 | 3.5898 | 3.5185 | 3.4134 | 3.4063 | 3.3565 |
> | KERN  | **4.2165** | **3.8650** | **3.7919** | **3.6724** | **3.5910** | **3.5503** | **3.4776** | **3.3707** | **3.3608** | **3.3080**
>
>
>
>
> **Q7:  This is partially true. Prior studies have indeed observed such trends, but the gap reported here is unusually large.**
>
> A7: **The performance gap is related to the experiment setting. ReMoE also has such a large performance gap, and the ReMoE' MoE baseline (with Softmax, achieving 39.67 in Table 2, the best MoE baseline) is only 0.3 better than Dense.**
>
> *The Performance in ReMoE Figure 5*
> | Method               | Loss |
> |----------------------|------------|
> |Dense (blue line)| About 2.05 loss
> | Softmax MoE w/o LB  (purple line)      | About 2.00 loss
> | ReMoE    w/o LB (brown line)      | About 1.90 loss|
>
>
> *The Table 9 Performance of ReMoE*
>
> | Method               | E |ARC-c | ARC-e | BoolQ | HellaSwag | LAMBADA | PIQA | RACE | Avg.
> |----------------------|------------|--------------|---|---|---|---|---|---|---|
> |Dense | | 19.45| 43.35| 54.40| 28.61| 31.09| 61.97| 28.52| 38.20|
> | MoE (Softmax)| 16| 20.90| 45.29| 46.36| 30.50| 33.22| 64.96| 28.33| 38.50 (increase 0.3, compared to Dense)|
> | ReMoE  | 16| 20.90| 49.28| 53.36| 30.85| 37.09| 65.83| 30.05| 41.05 (increase 2.55, compared to MoE )
> | MoE (Softmax)| 32|  19.54 | 47.35 | 52.29 | 31.12 | 35.63 | 64.25 | 28.23 | 39.77 (increase 1.57, compared to Dense) |
> | ReMoE  | 32| 20.56 | 48.11 | 59.54 | 31.42 | 37.84 | 65.18 | 28.42 | 41.58 (increase 1.81, compared to MoE )
> | MoE (Softmax)| 128|  20.99 | 47.69 | 56.73 | 32.00 | 36.62 | 65.67 | 28.04 | 41.10  (increase 2.90, compared to Dense) |
> | ReMoE  | 128| 19.97 | 51.05 | 56.97 | 32.40 | 37.92 | 66.70 | 29.86 | 42.12 (increase 1.02, compared to MoE )
>
> **Q8: Page 5, Line 218, "recent work has shown that replacing the Softmax with a Sigmoid router yields improved performance", please add citations.**
>
> A8: Thank you for the comment. We added the citation for clarification.
>
> **Q9: Typo and Writing**
>
> A9: Thank you for the precious comment. We have fixed the typos.
>
> **If the response addresses the concerns, could you consider improving the score? Thank you for the attention**

---

> ### Author Response · Authors · 2025-11-27
> **Kind Reminder of Discussion Period End**
>
> Dear Reviewer qEoZ,
>
> Hope this finds you well. We have addressed the concerns of insight, compared with the mentioned gelu, silu, and glu, the load balance setting, compared with other methods, the performance gap for softmax and sigmoid, and writing. **The discussion period will end soon. We truly believe the KERN will enhance the understanding of MoE and inspire future works of better MoE architecture. This work will  be important for MoE, as it enhances the understanding of MoE and the proposed method is zero-additional cost. Therefore, if the above response addresses the concerns, could you consider improving the score? Sincerely hope that the work could have your precious support. Thank you for the attention**

---

### Official Review · Reviewer_pDju · 2025-11-01

**Soundness:** 2
**Presentation:** 2
**Contribution:** 3
**Rating:** 2
**Confidence:** 3

**Summary:**

This paper revisits the routing mechanism of Mixture-of-Experts (MoE) from a statistical perspective. The authors draw an analogy between the MoE router and the Nadaraya–Watson (NW) kernel regression, showing that both can be viewed as weighted aggregations over functions. Motivated by this connection, the paper introduces KERN (Kernel-Inspired Router with Normalization) — an FFN-style router that replaces Softmax with ReLU activation and ℓ₂ normalization.
Extensive experiments across model sizes (125M–6.9B parameters), context lengths, and datasets (Arxiv, Books3, FineWeb-Edu, etc.) demonstrate consistent gains of KERN over Softmax, Sigmoid, and Tanh routers. The method adds no parameters or computational overhead, making it a practically attractive alternative for large-scale MoE systems.

**Strengths:**

This paper revisits the routing mechanism of Mixture-of-Experts (MoE) from a statistical perspective. The authors draw an analogy between the MoE router and the Nadaraya–Watson (NW) kernel regression, showing that both can be viewed as weighted aggregations over functions. Motivated by this connection, the paper introduces KERN (Kernel-Inspired Router with Normalization) — an FFN-style router that replaces Softmax with ReLU activation and ℓ₂ normalization.
Extensive experiments across model sizes (125M–6.9B parameters), context lengths, and datasets (Arxiv, Books3, FineWeb-Edu, etc.) demonstrate consistent gains of KERN over Softmax, Sigmoid, and Tanh routers. The method adds no parameters or computational overhead, making it a practically attractive alternative for large-scale MoE systems.

**Weaknesses:**

The central theoretical claim---that Mixture-of-Experts (MoE) routing is *equivalent* to Nadaraya–Watson (NW) kernel regression---is conceptually appealing but lacks mathematical rigor.

The presented argument only shows that the softmax weighting can *approximate* a kernel-weighted aggregation under certain parameterizations, not that the two models are formally or probabilistically equivalent.

In particular, NW regression requires a symmetric and positive-definite kernel function with explicit density normalization, while the softmax router relies on asymmetric learned logits and gradient-based optimization dynamics.

Hence, the connection is heuristic rather than theoretically grounded.

The paper would benefit from a clear statement of the conditions (e.g., scaling limits, normalization assumptions) under which this analogy approximately holds.

Overall, the claimed equivalence is oversold, weakening the theoretical soundness of the contribution.

**Questions:**

1. The paper claims a formal equivalence between softmax routing and Nadaraya–Watson kernel regression. Could the authors clarify whether this relationship is a *functional approximation* (similar weighting structure) or a *true probabilistic equivalence* under specific assumptions?

2. Under what mathematical conditions---such as temperature scaling, large-dimension limits, or specific kernel choices---can softmax routing converge to the NW estimator?

3. Given that the softmax router’s logits are learned through gradient descent rather than fixed distances, how does this affect the validity of interpreting it as a kernel regression model?

---

> ### Author Response · Authors · 2025-11-14
> **Response to Reviewer pDju (Part 1/3)**
>
> Dear Reviewer pDju,
>
> Thank you for the comment, we will address your concerns below.
>
>
> **Before all, it seems that it is better further clarify the relation between Nadaraya-Watson Kernel Regression, Mixture-of-Experts and Mixture-of-Experts Routing**
>
> **Nadaraya-Watson Kernel Regression**: $f_{NW}(x) = \sum_{i=1}^N \frac{K(x, x_i)}{\sum_{j=1}^N K(x, x_j)} y_i$
>
> **Mixture-of-Experts**:  $\text{MoE}(x) = \sum_{m=1}^M \hat{g}_m(x) E_m(x),$
>
> **Mixture-of-Experts Routing**: $g_m(x) = \frac{K(\langle w_m, x)}{\sum_{j=1}^M K(\langle w_j, x \rangle)}.$
>
> **We interprete the Mixture-of-Experts as the Nadaraya-Watson Kernel Regression, but not the Mixture-of-Expert Router. With the view of Nadaray-Watson Kernel Regression, the $y_i$ is the output of i-th expert, and the $\frac{K(x, x_i)}{\sum_{j=1}^N K(x, x_j)}$ is calcualted by the router.**
> More specifically, NW regression tries to predict by aggregating the label of other data. Similarly, for MoE, the output is the aggregation of experts' output. We regard experts' output as the label to be aggregated in NW regression.
>
>
> **Q1: The central theoretical claim---that Mixture-of-Experts (MoE) routing is equivalent to Nadaraya–Watson (NW) kernel regression is conceptually appealing but lacks mathematical rigor.**
>
> A1: The **Mixture-of-Experts is equivalent** to Nadaraya–Watson (NW) kernel regression, but the **Mixture-of-Experts routing is not equivalent**.
>
> **Nadaraya-Watson Kernel Regression**: $f_{NW}(x) = \sum_{i=1}^N \frac{K(x, x_i)}{\sum_{j=1}^N K(x, x_j)} y_i$
>
> **Mixture-of-Experts**:  $\text{MoE}(x) = \sum_{m=1}^M \hat{g}_m(x) E_m(x),$
>
> **Mixture-of-Experts Routing**: $g_m(x) = \frac{K(\langle w_m, x)}{\sum_{j=1}^M K(\langle w_j, x \rangle)}.$
>
> **MoE is the Nadaraya-Watson Kernel Regression, while the Mixture-of-Experts Routing is not. When  Nadaraya-Watson Kernel Regression with $K(x,w_i)=K(w_i,x)=e^{x \cdot w_i}$, it is the MoE with softmax router.**
>
> The Nadaraya–Watson (NW) kernel regression: calculate the similarity and then multiply the corresponding value, which is the same as the MoE: router logit could be regarded as the similarity, and the expert output is the value. After training, where all parameters are fixed, MoE is exactly doing NW regression, where router functions calculate the similarity scores and experts' outputs are labels $y_m$ to be aggregated. During the training process, we are updating some parameters in the router functions to construct and learn the suitable kernel functions.
>
> **Q2: The presented argument only shows that the softmax weighting can approximate a kernel-weighted aggregation under certain parameterizations, not that the two models are formally or probabilistically equivalent.**
>
> A2: **The MoE with softmax router is Nadaraya-Watson Kernel Regresion  when the $K(x,w_i)=K(w_i,x)=e^{x \cdot w_i}$. The softmax router itself is not the Nadaraya-Watson Kernel Regression, but the MoE  is the Nadaraya-Watson Kernel Regression. With different kernel functions of Nadaray-Watson Kernel, we could have different current variant of MoE, such as relu kernel for MoE with relu router (e.g. proposed by ReMoE), sigmoid kernel for MoE with sigmoid router (e.g. used in DeepSeek V3)**
>
> This work suggests that the MoE can be regarded as the Nadaraya–Watson (NW) kernel regression: calculate the similarity and then multiply the corresponding value. The Softmax Routing is a potential kernel function in the Nadaraya–Watson Kernel Regression, with $K(x,w_i)=K(w_i,x)=e^{x \cdot w_i}$.
>
> After training (at the inference stage), MoE is exactly doing NW regression, where router functions calculate the similarity scores and experts' outputs are labels $y_m$ to be aggregated. During the training process, we are updating some parameters in the router functions to construct and learn the suitable kernel functions.
>
> **Q3: In particular, NW regression requires a symmetric and positive-definite kernel function with explicit density normalization, while the softmax router relies on asymmetric learned logits and gradient-based optimization dynamics.**
>
> A3: **The MoE with softmax routering is the kernel  $K(x,w_i)=K(w_i,x)=e^{x \cdot w_i}$, which is actually symmetric.**
>
>
>
> The NW Kernel is **not necessarily be symmetric and positive-definite kernel function with Explicit Density Normalization**.
>
>
> * The kernel does not necessarily have to be symmetric. For example, $K(x,x_i)= e^{0.1(x \cdot x_i)}$ for $|x|<=|x_i|$ and $K(x,x_i)= e^{0.2(x \cdot x_i)}$$ for $|x|>|x_i|$ .
> * The kernel could be a uniform kernel function, which is not positive-definite. For example, $K(x,x_i)= 1/2$ for $|x-x_i|<=1$ and $K(x,x_i)= 0$ for $|x-x_i|>=1$.
> * The kernel is NOT necessary to have explicit normalization. The NW Kernel Regression has a denominator, which could be regarded as a normalization. Therefore, it is not necessary that the kernel function has explicit density normalization.

---

> > ### Author Response · Authors · 2025-11-14
> > **Response to Reviewer pDju (Part 2/3)**
> >
> > **Q4: Hence, the connection is heuristic rather than theoretically grounded. The paper would benefit from a clear statement of the conditions (e.g., scaling limits, normalization assumptions) under which this analogy approximately holds.**
> >
> > A4: **The only condition: find a kernel/method to calculate the similarity between x and $w_i$.** We have clearly presented the relation between Mixture-of-Experts and Nadaraya-Watson Kernel Regression. **For Nadaraya-Watson, we could choose any suitable kernel, such as softmax, sigmoid and so on. For MoE, we could choose any suitable, such as softmax or sigmoid**
> >
> > **Nadaraya-Watson Kernel Regression**: $f_{NW}(x) = \sum_{i=1}^N \frac{K(x, x_i)}{\sum_{j=1}^N K(x, x_j)} y_i$
> >
> > **Mixture-of-Experts**:  $\text{MoE}(x) = \sum_{m=1}^M \hat{g}_m(x) E_m(x),$
> >
> > **Mixture-of-Experts Routing**: $g_m(x) = \frac{K(\langle w_m, x)}{\sum_{j=1}^M K(\langle w_j, x \rangle)}.$
> >
> > **When  Nadaraya-Watson Kernel Regression with $K(x,w_i)=K(w_i,x)=e^{x \cdot w_i}$, it is the MoE with softmax router.**
> >
> >
> >
> > **Q5: The paper claims a formal equivalence between softmax routing and Nadaraya–Watson kernel regression. Could the authors clarify whether this relationship is a functional approximation (similar weighting structure) or a true probabilistic equivalence under specific assumptions?**
> >
> > A5: **It is a true probabilistic equivalence for MoE with softmax routing and Nadaraya-Watson Kernel with $K(x,w_i)=K(w_i,x)=e^{x \cdot w_i}$** We interprete the MoE process as Nadaraya-Watson Kernel  Regression: calculate similarity (finish by the router) and then multiply the value (value is the output of experts), whatever the kernel function is softmax kernel, sigmoid kernel or others.
> >
> > **Nadaraya-Watson Kernel Regression**: $f_{NW}(x) = \sum_{i=1}^N \frac{K(x, x_i)}{\sum_{j=1}^N K(x, x_j)} y_i$
> >
> > **Mixture-of-Experts**:  $\text{MoE}(x) = \sum_{m=1}^M \hat{g}_m(x) E_m(x),$
> >
> > **Mixture-of-Experts Routing**: $g_m(x) = \frac{K(\langle w_m, x)}{\sum_{j=1}^M K(\langle w_j, x \rangle)}.$
> >
> >
> > **The MoE with softmax routering is when the $K(x,x_i)=e^{x\cdot x_i} ($x_i$ is one of the router parameter, and $y_i$ is the expert output of i-th expert.**

---

> ### Author Response · Authors · 2025-11-14
> **Response to Reviewer pDju (Part 3/3)**
>
> **Q6: Under what mathematical conditions---such as temperature scaling, large-dimension limits, or specific kernel choices---can softmax routing converge to the NW estimator?**
>
> A6: **It is not related to the temperature scaling or large-dimension limits. The Nadaraya-Watson Estimator could have a different kernel function, while the MoE with softmax routing is the Nadaraya-Watson Kernel Regression with kernel $K(x,x_i)=e^{x\cdot x_i}$ for the following:**
>
> **Nadaraya-Watson Kernel Regression**: $f_{NW}(x) = \sum_{i=1}^N \frac{K(x, x_i)}{\sum_{j=1}^N K(x, x_j)} y_i$
>
> **Mixture-of-Experts**:  $\text{MoE}(x) = \sum_{m=1}^M \hat{g}_m(x) E_m(x),$
>
> **Mixture-of-Experts Routing**: $g_m(x) = \frac{K(\langle w_m, x)}{\sum_{j=1}^M K(\langle w_j, x \rangle)}.$
>
>
> where the $y_i$ is the expert output of i-th expert.
>
>
> **Q7: Given that the softmax router’s logits are learned through gradient descent rather than fixed distances, how does this affect the validity of interpreting it as a kernel regression model?**
>
> A7L **The router logit $x \cdot w_i$ is related to $w_i$, which is fixed after training.** We interpret the Mixture-of-Expert as Kernel Regression Model, but not softmax router. After training, given the same token embedding, the  $K(x,w_i)=e^{x\cdot w_i}$ is fixed, for $w_i$ is the learnable parameter of router.
>
> **Nadaraya-Watson Kernel Regression**: $f_{NW}(x) = \sum_{i=1}^N \frac{K(x, x_i)}{\sum_{j=1}^N K(x, x_j)} y_i$
>
> **Mixture-of-Experts**:  $\text{MoE}(x) = \sum_{m=1}^M \hat{g}_m(x) E_m(x),$
>
> **Mixture-of-Experts Routing**: $g_m(x) = \frac{K(\langle w_m, x)}{\sum_{j=1}^M K(\langle w_j, x \rangle)}.$
>
>
> For MoE, the $\frac{K(x, x_i)}{\sum_{j=1}^N K(x, x_j)}$ is calculated by the router, and the $y_i$ is the output of the $i-th$ expert.
>
> **We interprete the Mixture-of-Experts as the Nadaraya-Watson Kernel Regression**
> **We do not interpret the softmax router as Nadaraya-Watson Kernel Regression**
> **The softmax router is one of the potential kernel functions for $\frac{K(x, w_i)}{\sum_{j=1}^N K(x, w_j)}$, while $K(x,w_i)=e^{x\cdot w_i}$.**
>
> **After training, given the same token embedding, the  $K(x,w_i)=e^{x\cdot w_i}$ is fixed, for $w_i$ is the learnable parameter of router.**
>
>
> **If the response addresses your concerns, could you consider increasing the score?**

---

> > ### Author Response · Authors · 2025-11-17
> >
> > Dear Reviewer pDju,
> >
> > Thank you for the attention. **We have addressed the concerns in the rebuttal, including the relation between MoE and Nadaray-Watson Kernel Regression, the gradient analysis, softmax router, and so on.**
> >
> > Also, **We have highlighted that the softmax router is actually a symmetric kernel.**
> >
> > **If the response addresses your concerns, could you consider increasing the score?**

---

> ### Author Response · Authors · 2025-11-24
> **One Week Before Discussion End**
>
> Dear Reviewer pDju,
>
> The response was presented about two weeks ago, and there is about one week left for the discussion. **What else could we do to help consider improving the score? If there are no further concerns, could you consider improving the score? Thank you for the attention and engagement, and wish you a good day.**

---

> > ### Author Response · Authors · 2025-11-27
> > **The Dicussion will End in 3 Working Days**
> >
> > Dear Reviewer pDju,
> >
> > Hope this finds you well. **As the rebuttal will end in three working days, if the response addresses the concerns, could you consider to increase the score? Thank you for the attention**

---

> ### Author Response · Authors · 2025-11-27
> **Kind Reminder of Discussion Period End**
>
> Dear pDju,
>
> Hope this finds you well. We have addressed the concerns of the connection between MoE and Nadarya-Watson Kernel Regression, the kernel requirement. **The discussion period will end soon. For this work, we connect the MoE and FFN with Nadaray-Watson Kernel Regression, and then bring the successful architecture of FFN to MoE. This work  will improve the understanding of MoE. As it is zero additional cost, it will be easy to deploy in the real application. Of there is any questions, please let us know. Sincerely hope that this work could have your precious support. Thank you for the attention**

---

### Official Review · Reviewer_dQSC · 2025-11-04

**Soundness:** 3
**Presentation:** 3
**Contribution:** 2
**Rating:** 6
**Confidence:** 3

**Summary:**

This paper connects MoE routers to Nadaraya-Watson regression. The authors then propose KERN, a zero-cost FFN-style router using ReLU activation and ℓ2-normalization. Experiments on language modelling and NLP tasks demonstrate KERN's effectiveness.

**Strengths:**

- The authors motivates various design decisions thoroughly.
-  This work connects kernel method perspective to MoE router design.

**Weaknesses:**

- Improvement is usually marginal.
- It's a pity that the connection to Nadaraya-Watson estimator isn't well explored. It's not obvious why the router design is inspired by this perspective, they don't have much shared element, and somehow we don't see any attempt at adopting the traditional formula as a router. I think the paper will be a lot more interesting if it shows variants like Gaussian kernel.

**Questions:**

- In "Advantages of the KERN router function", why does KERN help with gradient vanishing? Sigmoid and Softmax may still have some gradient even if the input is at small value, but RELU function in KERN directly cutoff the gradient.

---

> ### Author Response · Authors · 2025-11-14
> **Response to Reviewer dQSC (Part 1/2)**
>
> Dear Reviewer dQSC,
>
> Thank you for your support of our work. We will address the concerns below.
>
> **Q1: Improvement is usually marginal.**
>
> A1: **Our method does not introduce additional computational cost, but achieves about 0.06 loss reduction (compared to softmax router), which is significant. For example, the recent gated attention (NeurIPS 2025 Oral) [1] only reduces 0.02 loss, while including additional parameters and computational cost.**
>
> * **We also challenge a common wrong belief** in Mixture-of-Experts: previous works believe that the more activated parameters, the better the performance
> * However, as **validated in Appendix J**, we find that adding ReLU in router logit actually improves the performance, just like the FFN
>
> **Q2: It's not obvious why the router design is inspired by this perspective**
>
> A2: **We connect the FFN and MoE with the Nadaraya-Watson kernel Regression: calculate similarity and then multiply the value.
> More specifically, we can observe the similar structure of NW regression (in Equation (1)) and MoE aggregation (in Equation (5)). We note that NW regression is the aggregation of all data, where the weights depend on the similarity measured by kernel functions. MoE's output depends on the aggregation of experts' outputs. Therefore, the MoE shares a structure with NW regression. Based on this insight, we can interpret MoE aggregation from a more general NW regression perspective, not just the probability distribution of softmax. We also observe that FFN is a special case of NW regression, as illustrated in Equation (3).
> Then, we design the MoE router following the current widely adopted FFN structure, which is ReLU(Norm(MLP(x)).**
>
> **Nadaraya-Watson Kernel Regression**: $f_{NW}(x) = \sum_{i=1}^N \frac{K(x, x_i)}{\sum_{j=1}^N K(x, x_j)} y_i$
>
>
>
> **Mixture-of-Experts**:  $\text{MoE}(x) = \sum_{m=1}^M \hat{g}_m(x) E_m(x),$
>
> **Mixture-of-Experts Routing**: $g_m(x) = \frac{K(\langle w_m, x\rangle)}{\sum_{j=1}^M K(\langle w_j, x \rangle)}.$
>
> **FFN**:  $\text{FFN}(x) = \sum_{m=1}^M \phi(Norm(xW_1)_m)(W_2)_m$
>
> **MoE with KERN:** $\text{MoE}(x) = \sum_{m=1}^M \phi(Norm(xW)_m) E_m(x),$ where $\phi$ is ReLU.
>
>
> **Q3:  I think the paper will be a lot more interesting if it shows variants like Gaussian kernel**
>
> A3: The following is the gaussian kernel for $K( w_m, x)=\exp(-\|w_m-x\|^2/2\sigma^2)$.
>
> *The performance with loss metrics*
> |MoE Router |  5K | 10K | 15K | 20K | 25K | 30K | 35K | 40K | 45K | 50K |
> |---|----|-----|-----|-----|-----|-----|-----|-----|-----|-----|
> | Softmax  |4.3011 | 3.9318 | 3.8586 | 3.7394 | 3.6594 | 3.6185 | 3.5507 | 3.4459 | 3.4387 | 3.3882
> | Sigmoid  | 4.2520 | 3.8801 | 3.8051 | 3.6814 | 3.6011 | 3.5580 | 3.4869 | 3.3806 | 3.3706 | 3.3206
> | Tanh  | 4.2531 | 3.8915 | 3.8206 | 3.7017 | 3.6202 | 3.5764 | 3.5048 | 3.3978 | 3.3903 | 3.3388
> | Gaussian  | 4.2906 | 3.9058 | 3.8340 | 3.7121 | 3.6328 | 3.5909 | 3.5209 | 3.4110 | 3.4042 | 3.3531
> | KERN  |  **4.2165** | **3.8650** | **3.7919** | **3.6724** | **3.5910** | **3.5503** | **3.4776** | **3.3707** | **3.3608** | **3.3080**

---

> ### Author Response · Authors · 2025-11-14
> **Response to Reviewer dQSC (Part 2/2)**
>
> **Q4: In "Advantages of the KERN router function", why does KERN help with gradient vanishing? Sigmoid and Softmax may still have some gradient even if the input is at small value, but RELU function in KERN directly cutoff the gradient.**
>
> A4: **There is L2 normalization and the MoE is sparse (usually activation ratio is smaller than 50%), and the ReLU only cutoff value when the activation ratio is larger than 50% (Appendix J).** Therefore, even if the same value is cut off, it still has the gradient. The following is the gradient Jacobian Matrix for L2 normalization:
>
> $J = \frac{I}{\|x\|_2} - \frac{x x^T}{\|x\|_2^3}.$
>
> **Therefore, even if there is only one element that is selected, there is still a gradient on the cutoff position.** However, the softmax and sigmoid will face gradient vanishing when the value is very large or very small.
>
>
> ```
> Example: input [-2 -3 -1 2 3 10] :
> KERN (L2Norm): [-0.177  -0.266 -0.088  0.177   0.266  0.887]
> Sigmoid: [0.119 0.0474 0.268 0.880 0.952 0.999]
> ###The router logit will not have a gradient for $w_6$
> Softmax: [6.136e-06 2.257e-06 1.668e-05 3.350e-04 9.107e-04 9.987e-01]
> ######The graient is almost vanish for $w_1$ to $w_6$
>
> ReLU & choose top-2 for KERN: [0 0 0 0 0.266  0.887]
> #### $w_1$ to $w_6$ all have gradient because of L2 normalization.
> #### Note that whether there is ReLU or not, the choice of top-2 will not change the final result because usually the ReLU only masks 50%, while top-2 masks 4/6=66.7%.
> #### When the top-k for K/M>50%, adding ReLU may improve the performance, which is similar to FFN (Appendix J).
>
> choose top-2 for Sigmoid: [0 0 0 0 0.952 0.999]
> #### the sigmoid will NOT have gradient for $w_1$ to $w_4$ because only 2 values are selected. And $w_6$ also does not have a gradient that vanishes because 0.999 is too large.
>
> choose top-2 for Softmax: [0 0 0 0 9.107e-04 9.987e-01]
> ### the softmax will NOT have a gradient for $w_1$ to $w_6$ because one value is very large.
> ```
>
>
>
>
>
> Reference:
>
> [1] Qiu, Z., Wang, Z., Zheng, B., Huang, Z., Wen, K., Yang, S., ... & Lin, J. (2025). Gated Attention for Large Language Models: Non-linearity, Sparsity, and Attention-Sink-Free. arXiv preprint arXiv:2505.06708.
>
>
> **If the response addresses your concerns, could you consider increasing the score?**

---

> ### Author Response · Authors · 2025-11-18
>
> Dear Reviewer dQSC,
>
> Thank you for your support of our work.  We explain that the improvement is actually significant, the motivation of the design (connect FFN and MoE with Nadaray-Watson Kernel Regression; Bring FFN success to MoE), the performance with the Gaussian kernel, and the gradient analysis. **What else could we do to help consider improving the score? If there are no further concerns, could you consider improving the score?**

---

> > ### Comment · Reviewer_dQSC · 2025-11-18
> >
> > Thanks a lot for the explanation and results. I'm convinced.

---

> > > ### Author Response · Authors · 2025-11-19
> > > **Response to Reviewer dQSC**
> > >
> > > Dear Reviewer dQSC,
> > >
> > > Thank you very much for the support, which helps a lot. Again, thank you for the attention, and wish you a good day.

---

### Author Response · Authors · 2025-11-14
**Response to All Reviewers**

Dear Area Chair and Reviewers,

Thank you for the attention to this work. We have responded to the question of reviewers. And the following is the response to the common questions.


**Q1: The relation between Nadaraya-Watson Kernel Regression, FFN, and MoE ( Reviewer dQSC, Reviewer pDju, Reviewer qEoZ, Reviewer XdAq)**

A1: **We connect the FFN and MoE with Nadaraya-Watson Kernel Regression, and then we design the MoE as FFN**

**Nadaraya-Watson Kernel Regression**: $f_{NW}(x) = \sum_{i=1}^N \frac{K(x, x_i)}{\sum_{j=1}^N K(x, x_j)} y_i$



**Mixture-of-Experts**:  $\text{MoE}(x) = \sum_{m=1}^M \hat{g}_m(x) E_m(x),$

**Mixture-of-Experts Routing**: $g_m(x) = \frac{K(\langle w_m, x\rangle)}{\sum_{j=1}^M K(\langle w_j, x \rangle)}.$

**FFN**:  $\text{FFN}(x) = \sum_{m=1}^M \phi(Norm(xW_1)_m)(W_2)_m$

**MoE with KERN:** $\text{MoE}(x) = \sum_{m=1}^M \phi(Norm(xW)_m) E_m(x),$

The MoE is the Nadaraya-Watson Kernel Regression with $K(x,w_i)=K(w_i,x)=e^{x \cdot w_i}$

**Q2: The gradient related to KERN, Sigmoid and Softmax (Reviewer dQSC, Reviewer qEoZ)**

A2: **There is L2 normalization and the MoE is sparse (usually activation ratio is smaller than 50%), and the ReLU cutoff value when the activation ratio is larger than 50% (Appendix J).** Therefore, even if some value is cut off, it still has the gradient. The following is the Jacobian Matrix:

$J = \frac{I}{\|x\|_2} - \frac{x x^T}{\|x\|_2^3}.$

**Therefore, even if there is only one element that is selected, there is still a gradient on the cutoff position of KERN.** However, the softmax and sigmoid will face gradient vanishing when the value is very large or very small. For example, we observe that the softmax value and sigmoid value could achieve relatively large values (e.g,. more than 0.99).





```
Example: input [-2 -3 -1 2 3 10] :
KERN (L2Norm): [-0.177  -0.266 -0.088  0.177   0.266  0.887]
Sigmoid: [0.119 0.0474 0.268 0.880 0.952 0.999]
###The router logit will not have a gradient for $w_6$
Softmax: [6.136e-06 2.257e-06 1.668e-05 3.350e-04 9.107e-04 9.987e-01]
######The graient is almost vanish for $w_1$ to $w_6$

ReLU & choose top-2 for KERN: [0 0 0 0 0.266  0.887]
#### Note that whether there is ReLU or not, the choice of top-2 will not change the final result because usually the ReLU only masks 50%, while top-2 masks 4/6=66.7%. When the top-k for K/M>50%, adding ReLU may improve the performance, which is similar to FFN (Appendix J).
choose top-2 for Sigmoid: [0 0 0 0 0.952 0.999]
#### the sigmoid will NOT have gradient for $w_1$ to $w_4$ because only 2 values are selected. And $w_6$ also does not have a gradient that vanishes because 0.999 is too large.
choose top-2 for Softmax: [0 0 0 0 9.107e-04 9.987e-01]
### the softmax will NOT have a gradient for $w_1$ to $w_6$ because one value is very large.
```

---

### Author Response · Authors · 2025-11-28
**Kind Reminder of the Discussion Period End**

Dear Reviewers,

Hope this finds you well. As the discussion period will end soon, we sincerely hope that you could check whether you might have a chance to respond to our clarification. We fully understand that you must be extremely busy, but your feedback would be valuable for us to improve and better address your concerns before the final submission.

**We deeply appreciate your time and effort in reviewing our paper, and thank you again for your thoughtful comments and suggestions.**

---

### Author Response · Authors · 2025-11-28
**Response to the ICLR Updated Policy**

Dear Area Chair and Program Chair,

According to the new policy
* **It will be re-assigned each paper to a new area chair.**
   * **If the Re-Assignment is finished and the New Area Chair comes, please let us know. If there are any questions or concerns about this work, we will respond immediately. Thank you for the attention.**
* **It will be reverting reviews and scores to their state before the discussion period.**
   * Thank you for the arrangement. **We would like to highlight that the Reviewer dQSC increased the score from 6 to 8 on 18/19 Nov, which is far from the 27 Nov when the data was leaked.**

**We are really grateful for the precious effort and attention of the two area chairs of this work. The following is the core contribution**
* We connect the FFN and MoE with Nadaraya-Watson Kernel Regression, which is well discussed in the Method Part.
* We bring the success of FFN architecture to MoE, with the proposed method KERN in the Method Part
* We conduct extensive experiments to validate the proposed method, KERN in the Experiment Part.
* We find that the more activated parameter of MoE does not always improve the performance, just like FFN (e.g., FFN with ReLU activation 50% parameter to have better performance), which is validated in Appendix J.
* We believe that this work will enhance the understanding of MoE, and inspire future work to develop a better architecture of MoE.

**Again, we really appreciate the attention and effort of the discussion arrangement of the two area chairs, and we are grateful for the time and effort of the discussion of the reviewers. We sincerely hope that the community could be better in the future**

---

### Meta-Review · Area_Chair_9hJb · 2026-01-11

**Summary:**

- Marginal gains, unclear motivation from the kernel perspective, and gradient behavior.
- More comparisons, details on load balancing, clarification of the sigmoid-softmax performance gap;
- Deeper discussion of the relation between activated parameter number and performance.

**Reviewer Concerns:**

Concerns addressed:
- Theoretical grounding of the NW–MoE connection was addressed through explicit formulations showing how MoE aggregation corresponds to Nadaraya–Watson regression after training.
- Why KERN helps with gradient behavior compared to Softmax/Sigmoid was addressed with Jacobian analysis, normalization effects, sparsity regimes, and concrete numerical examples.
- Marginal empirical improvements was contextualized relative to zero additional cost.
- The lack of kernel variants (e.g., Gaussian) was addressed by additional experiments showing Gaussian kernel baselines.

Concerns remain:
- None from the reliable reviewers, as the authors provided responses to all raised questions.

**Reviewer Scores:**

Reliable reviewers (dQSC,XdAq)  are likely to keep the scores above acceptance threshold.

---

### Decision · Program_Chairs · 2026-01-26

Accept (Poster)